# Casein kinase 1a mediates a two-step subunit remodeling mechanism to regulate the FRQ-FRH circadian clock complex

Carolin Schunke[1], Bianca Ruppert[1], Linda Lauinger[2], Sabine Schultz[1], Axel C. R. Diernfellner [1] & Michael Brunner [1] ✉

The circadian clock of *Neurospora* operates through a negative feedback loop in which FREQUENCY (FRQ), along with FRQ-interacting RNA helicase (FRH) and casein kinase 1a (CK1a), inhibits its transcriptional activator, WHITE COLLAR COMPLEX (WCC), via phosphorylation. CK1a, anchored to FRQ, hyperphosphorylates FRQ at its intrinsically disordered regions in a slow, temperature-independent manner, forming a module suited for molecular timekeeping. However, the molecular processes triggered by FRQ's hyperphosphorylation have remained unclear. Here we show that FRH, the folded binding partner of disordered FRQ, decodes FRQ's time-dependent phosphorylation state by triggering a two-step remodeling of the FRQ-FRH complex: initially, two FRH molecules bind a FRQ dimer, keeping it inactive by blocking its interaction with WCC. Slow phosphorylation eventually triggers the dissociation of one FRH, thereby activating the complex by exposing a WCC-binding site. Due to the slow and stochastic nature of phosphorylation, the release of the second FRH occurs with a delay, promoting nuclear export and subsequent degradation of FRQ. This ensures precise activation and inactivation of FRQ and positions FRH as a hub for decoding temporal phosphorylation information.

Circadian clocks are fascinating molecular devices capable of accurately measuring time and orchestrating the physiology and behavior of organisms in sync with the 24-h environmental day-night cycle. The circadian clock of the filamentous fungus *Neurospora crassa* is based on a delayed transcriptional-translational negative feedback loop: The heterodimeric circadian transcription factor White Collar Complex (WCC), composed of a White Collar-1 (WC-1) and WC-2 subunit, activates transcription of the circadian clock gene *frequency* (*frq*)[1–3]. The FRQ protein, 85% of which consists of intrinsically disordered regions (IDRs)[2,4,5], dimerizes via a coiled-coil in its N-terminal region[6]. FRQ-Interacting RNA-helicase, FRH, binds to the C-terminal third of FRQ and protects FRQ from premature degradation[7]. FRH is an orthologue of the conserved MTR-4 helicase[7], which has essential functions in various aspects of RNA metabolism[8]. Helicase activity of FRH is not

required for its moonlighting function in the circadian clock[5]. Casein kinase 1a (CK1a) binds to the central part of FRQ via the spatially separated FRQ-CK1-interaction Domains 1 and 2, FCD1 and FCD2[9,10]. The FRQ-FRH-CK1a complex (FFC) interacts weakly and dynamically with WCC and facilitates its inactivation through phosphorylation by CK1a[11]. In the course of a circadian period, FRQ is progressively hyperphosphorylated leading to its inactivation and degradation[12–14]. More than 100 phosphorylation sites in FRQ have been identified[15,16], the vast majority located in its IDRs[15,17,18]. Hyperphosphorylation is associated with circadian timekeeping and triggers a poorly characterized conformational change in the FFC[10,19]. FCD-bound CK1a is sufficient to phosphorylate FRQ on a circadian timescale and in a temperature-insensitive manner, and thus the FFC constitutes a molecular module suited to measure time[18,20–22]. Complete

[1]Heidelberg University Biochemistry Center, Heidelberg, Germany. [2]Umlaut.bio GmbH, Heidelberg, Germany.
✉e-mail: michael.brunner@bzh.uni-heidelberg.de

compensation of the clock at high temperature requires additionally phosphorylation of FRQ by casein kinase 2[23].

Despite extensive knowledge of the components of the core clock and their interactions, it is still unclear how time is measured at the molecular level, which is the essential function of a circadian clock. This also applies to the circadian clocks of metazoans, vertebrates, and invertebrates.

For example, it is not yet known how FRH protects FRQ from degradation, or whether this is its sole function. Furthermore, it is not known when and how the FFC interacts with WCC. Most significantly, we do not understand what kind of conformational change could be triggered by the partially redundant multisite phosphorylation of IDRs in FRQ and how this allows the FFC to perform its functions at the right time.

Answering these questions in *Neurospora* is a technical challenge and in many cases, not possible. For example, homologues of CK1a and FRH in yeast and mammals have essential functions outside the clock[24–28] and Neurospora knock-out strains are not available. FRQ and WC-1, without their binding partners, FRH and WC-2[5,29,30], respectively, are not expressed at a sufficient level to pursue complex biochemical analyses. Furthermore, *Neurospora* grows as a syncytium and forms dense hyphal networks with many nuclei that are rapidly transported along the hyphae[31], making live-cell microscopy challenging. In addition, the WCC is a light receptor that resets the circadian clock in response to blue light stimuli[32,33]. Therefore, analyzing the subcellular distribution and dynamics of clock proteins by imaging in living cells is technically challenging.

To overcome these challenges, we opted to heterologously express fluorescently tagged *Neurospora* clock proteins in U2OStx cells, enabling us to analyze their interactions and subcellular dynamics using a live cell imaging system. This straightforward approach enabled us to explore processes that are challenging or impossible to study in a cell-free in vitro system or in *Neurospora*. Based on these observations, we were then able to design specific experiments for validation in *Neurospora* that we might not have originally considered.

The approach yielded unexpected insights into the timekeeping mechanisms of the core clock. Our data reveal that the phosphorylation state of FRQ's IDRs is decoded through its interaction with its folded partner, FRH. FRH blocks the nuclear export signal (NES) of FRQ and the binding site for WCC, initially allowing accumulation of inactive inhibitor complexes that are activated with a delay by slow phosphorylation of FRQ. Phosphorylation eventually triggers the sequential stepwise release of FRH molecules. FFC subunit remodeling regulates FRQ's delayed interaction with WCC, as well as its nuclear export and degradation.

## Results

### FRQ contains three nuclear localization signals

To analyze the subcellular localization of FRQ, we constructed plasmids expressing full-length and truncated FRQ versions with either an N-terminal mKate2 (mK2) or a C-terminal mNeonGreen (mNG) moiety under control of a tetracycline-inducible CMV promoter (Fig. 1A and Supplementary Fig. 1A). The proteins were heterologously expressed in U2OStx cells via transient transfection, and the fluorescently tagged proteins were analyzed using an Incucyte live-cell imaging system. Overexpressed mK2-FRQ and FRQ-mNG accumulated in nuclear foci (Fig. 1B left panels, 1C and Supplementary Fig. 1B). FRQ[6B2] denotes a DHF to AAA substitution of amino acid residues 774–776 in FRQ, which is part of a larger FRH binding site, FFD, and abolishes binding of FRH[29,34], and FRQ[9] is a protein truncated after aa 662[35], entirely lacking the FRH binding site (Fig. 1A). mK2-FRQ[6B2] formed nuclear foci like full-length mK2-FRQ and mK2-FRQ[9] formed nuclear foci, which were slightly smaller (Fig. 1B). The data suggest that FRQ has the potential to self-interact and forms thermodynamically stable foci when overexpressed.

FRQ contains three polybasic sequences that could function as nuclear localization signal, NLS (Fig. 1A and Supplementary Fig. 1C). We expressed mK2-tagged fragments of FRQ, each containing one of the three polybasic sequences, mK2-FRQ[NLS1] (aa1–412), mK2-FRQ[NLS2] (aa557–638), and mK2-FRQ[C-term] (aa630–989) (Fig. 1A). The proteins localized to the nucleus, although a substantial portion of mK2-FRQ[C-term] was detected in the cytosol. In contrast, mK2-FRQ[C-term-NLS] (aa720–989), a C-terminal fragment lacking the polybasic sequence, accumulated in the cytosol (Fig. 1A, D). These data suggest that FRQ contains three NLSs, henceforth referred to as NLS1, NLS2, and NLS3.

### Impact of NLSs on FRQ function in *Neurospora*

We then created *Neurospora* strains expressing FRQ versions with mutated NLSs (Supplementary Fig. 1C) under control of the native *frq* promoter, alongside a control strain expressing wild-type (WT). These FRQ-mutant strains also expressed a *frq-lucP* reporter to enable analysis of circadian rhythms in vivo through bioluminescence recordings. The strains, grown in 96-well plates, were exposed to 12 h light, 12 h dark, and 12 h light and then transferred to the dark. The control strain, expressing WT FRQ, displayed a robust bioluminescence rhythm in constant darkness (Fig. 2A). Strains expressing FRQ[mNLS1] (NLS1: RRKKR to RQKKQ) displayed a WT period which slowly damped (Fig. 2B), suggesting that FRQ[mNLS1] must be a dimer in contrast to FRQ[ΔNLS1], which is a monomer and has lost all clock functions[36]. This finding suggests that the arrhythmic phenotype of FRQ[ΔNLS1] is likely due to the disrupted FRQ dimerization.

Strains expressing FRQ[mNLS2] (NLS2: RRKKRK to RGQERK) displayed a long period rhythm and strains expressing FRQ[mNLS3] (NLS3: RRKRR to RGQGR) displayed an early circadian phase and a short period rhythm damping slightly after several days in the dark (Fig. 2C, D). A triple mutant, FRQ[mNLS1/2/3], displayed a long period rhythm that damped rapidly in constant darkness (Fig. 2E). The data indicate that the NLSs are crucial for robust circadian oscillations, although some residual clock functions are retained in their absence. Subcellular fractionation revealed that all FRQ-NLS mutants, including FRQ[mNLS1/2/3], displayed nuclear-cytoplasmic distribution similar to WT FRQ (Supplementary Fig. 1D). In *Neurospora*, FRQ shuttles between the nucleus and the cytosol[37]. Although it is enriched in the nucleus[4,38], the majority of FRQ resides in the cytosol. This distribution is in part a consequence of the low nuclear-to-cytoplasmic volume ratio in filamentous fungi. However, since FRQ[9] was confined to the nucleus (Supplementary Fig. 1D), the C-terminal third of FRQ likely harbors a NES that governs FRQ's subcellular distribution. Because FRQ[mNLS1/2/3] is exclusively cytosolic in U2OStx cells, our data suggest that FRQ[mNLS1/2/3] must employ an additional mechanism to enter the nuclei in *Neurospora*, potentially through interaction with FRH.

### FRH binding governs FRQ self-interaction and localization

To analyze the subcellular localization of FRQ[mNLS1/2/3] alone and the influence of FRH, we generated plasmids encoding mK2-FRQ[mNLS1/2/3] and FRH C-terminally tagged with mNG, and expressed them separately or together in U2OStx cells. When expressed alone, FRH-mNG was evenly distributed in the nucleus and mK2-FRQ[mNLS1/2/3] accumulated in cytosolic foci, indicating the absence of additional NLSs (Supplementary Fig. 2A). Upon co-expression, the phenotype depended on the ratio of the transfected plasmids (Fig. 3A, B). At a DNA ratio of 16.5 ng FRH-mNG to 50 ng mK2-FRQ[mNLS1/2/3] (hereafter referred to as sub-saturating FRH levels), FRH-mNG co-localized with mK2-FRQ[mNLS1/2/3] in cytosolic foci (Fig. 3A). At saturating FRH levels (50 ng: 50 ng), however, both proteins were evenly distributed in the nucleus (Fig. 3B). These results indicate that the proteins interact at both expression ratios. Importantly, saturating amounts of FRH-mNG impaired mK2-FRQ[mNLS1/2/3] foci formation and promoted nuclear translocation of the NLS-deficient protein. Since each FRQ monomer has a single FRH binding site, our data strongly suggest that cytosolic

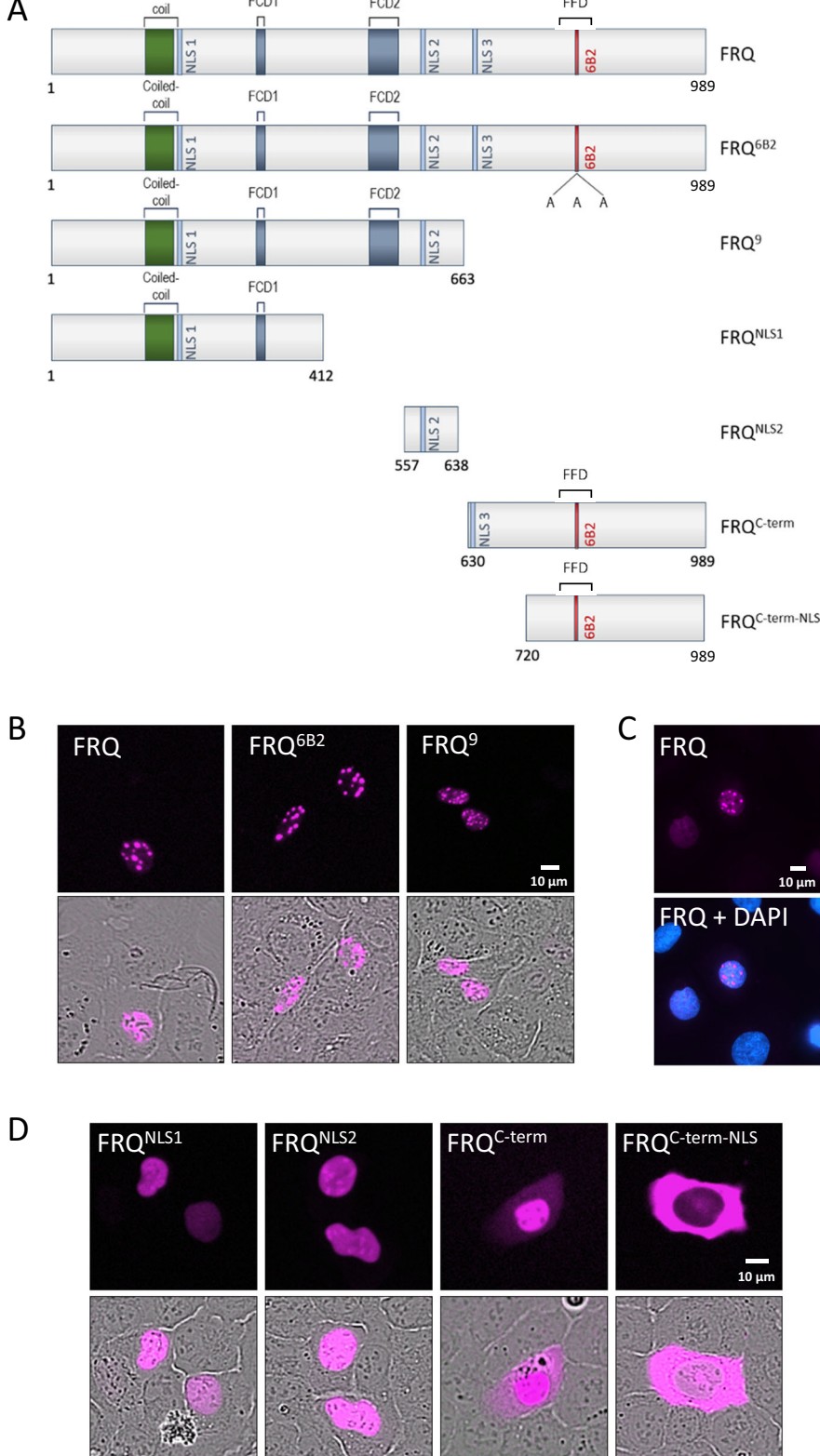

**Fig. 1 | FRQ contains three NLSs. A** Schematic of FRQ constructs. The coiled-coil dimerization domain, the predicted NLSs, the FRQ-CK1a interaction domains (FCD) 1 and 2, as well as the 6B2 region required for FRH binding to the FRQ-FRH binding domain (FFD)[9,10,29] are indicated. The FRQ[6B2] mutant carries an alanyl substitution of residues 774–776, DHF to AAA, which are part of the FFD[29,34]. The proteins were tagged either N-terminally with mK2 or C-terminally with mNG. **B** Transient expression of the indicated mK2-tagged FRQ variants in U2OStx cells, monitored by Incucyte live-cell microscopy (*n* = 3). Lower panels: overlays with phase contrast images to visualize nuclei (Incucyte cannot detect DAPI). **C** Wide-field fluorescence microscopy of mK2-FRQ with DAPI staining to confirm nuclear localization (*n* = 2). **D** Incucyte microscopy of FRQ NLS mutants (*n* = 3). Panels show subcellular expression patterns present in >90% of cells.

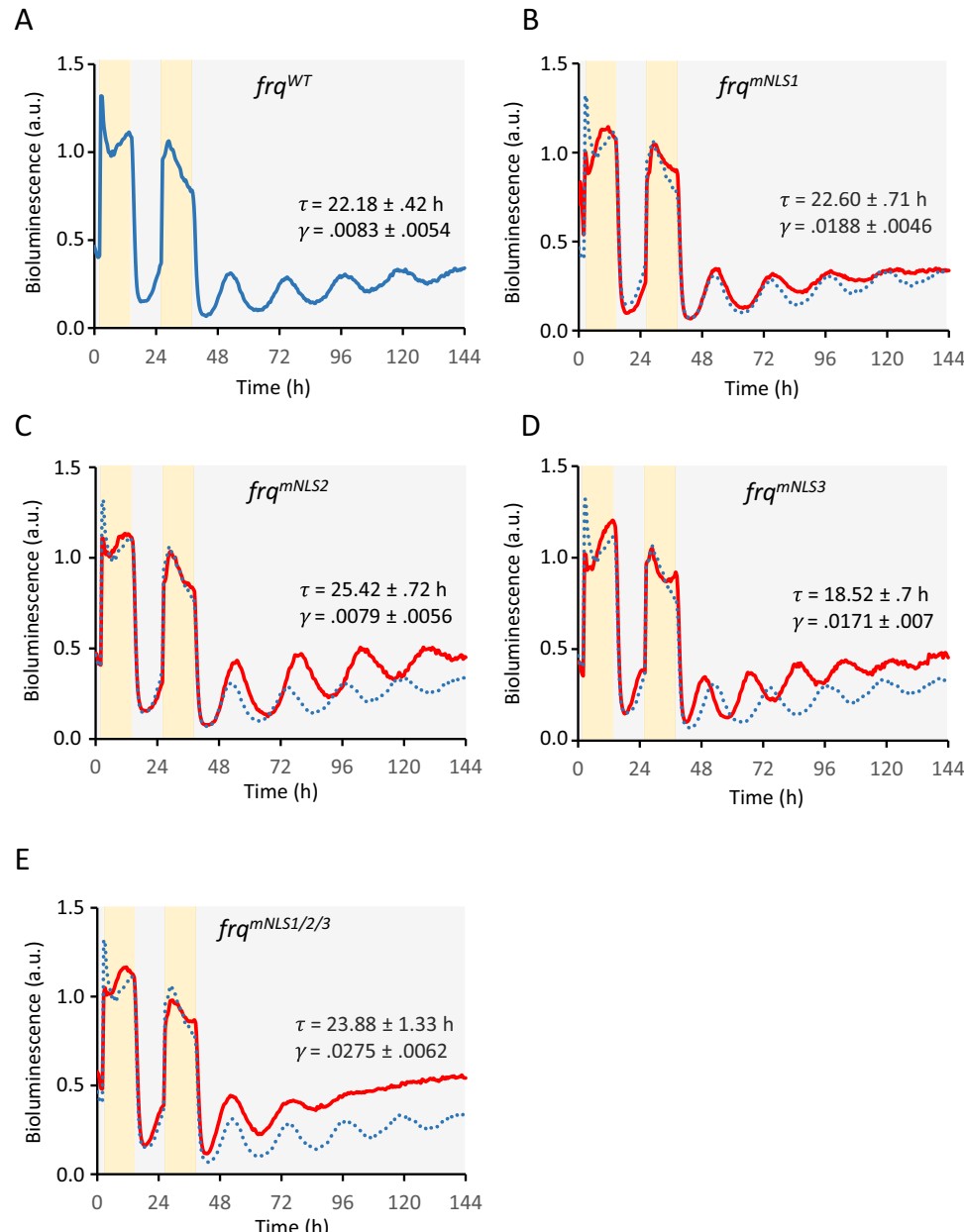

**Fig. 2 | Impact of FRQ's NLSs on the circadian clock. A–E** Bioluminescence recordings of *Neurospora frq-lucP* reporter strains expressing the indicated *frq* alleles. Calculated period lengths ($\tau \pm$ SD) and damping coefficients ($\gamma \pm$ SD) are shown. Results shown are from at least two independent experiments for each construct with a total of clones specified. **A** WT *frq*. The trajectory is the normalized average of 31 replicates from 4 independent clones. Also shown as dotted blue line for comparison in (**B–E**). **B** *frq^mNSL1*. 27 replicates, 7 clones. **C** *frq^mNLS2*. 15 replicates, 3 clones. **D** *frq^mNLS3*. 16 replicates, 3 clones. **E** *frq^mNLS1/2/3*. 40 replicates, 11 clones.

species of FRH-mNG represent heterotrimeric complexes of one FRH-mNG molecule bound to mK2-FRQ^mNLS1/2/3 dimers. At saturating FRH levels, binding of two FRH-mNG molecules interferes with foci formation and drives nuclear import of the heterotetrameric complex. The data further suggest that mK2-FRQ^mNLS1/2/3 must contain a NES that competes with FRH-mediated import.

mK2-FRQ, which possesses three NLSs forms nuclear foci on its own (see Fig. 1B). When co-expressed with sub-saturating FRH-mNG levels, both proteins accumulated in nuclear foci, whereas saturating FRH-mNG levels disrupted foci formation (Fig. 3C, D).

We then expressed mK2-FRQ^6B2 together with saturating FRH-mNG. mK2-FRQ^6B2 accumulated in nuclear foci, which did not contain FRH-mNG; the latter was homogenously dispersed in the nucleus (Fig. 3E). This confirms that the 6B2 mutation disrupts FRQ-FRH binding[29].

The C-terminally truncated protein mK2-FRQ^9 also accumulated in nuclear foci regardless of FRH-mNG levels (Fig. 3F). Finally, when we co-expressed mK2-FRQ^C-term, which contains NLS3, with saturating FRH-mNG (Fig. 3G), both proteins localized evenly throughout the nucleus, as expected. However, comparison with mK2-FRQ^C-term expressed alone (Fig. 1D) revealed more pronounced nuclear localization in the presence of FRH-mNG, suggesting that FRH enhances nuclear accumulation of this fragment. Quantifications are summarized in Supplementary Table 1.

Together, these findings indicate that tight anchoring of FRH to its binding domain (FFD) in the 6B2 region of FRQ prevents FRQ accumulation in nuclear foci, a process that requires additional FRQ regions outside the FRH anchoring site. In the absence of site-specific tight anchoring, FRH does not interact with FRQ. Anchoring thus increases the local FRH concentration, which may facilitate weak interactions

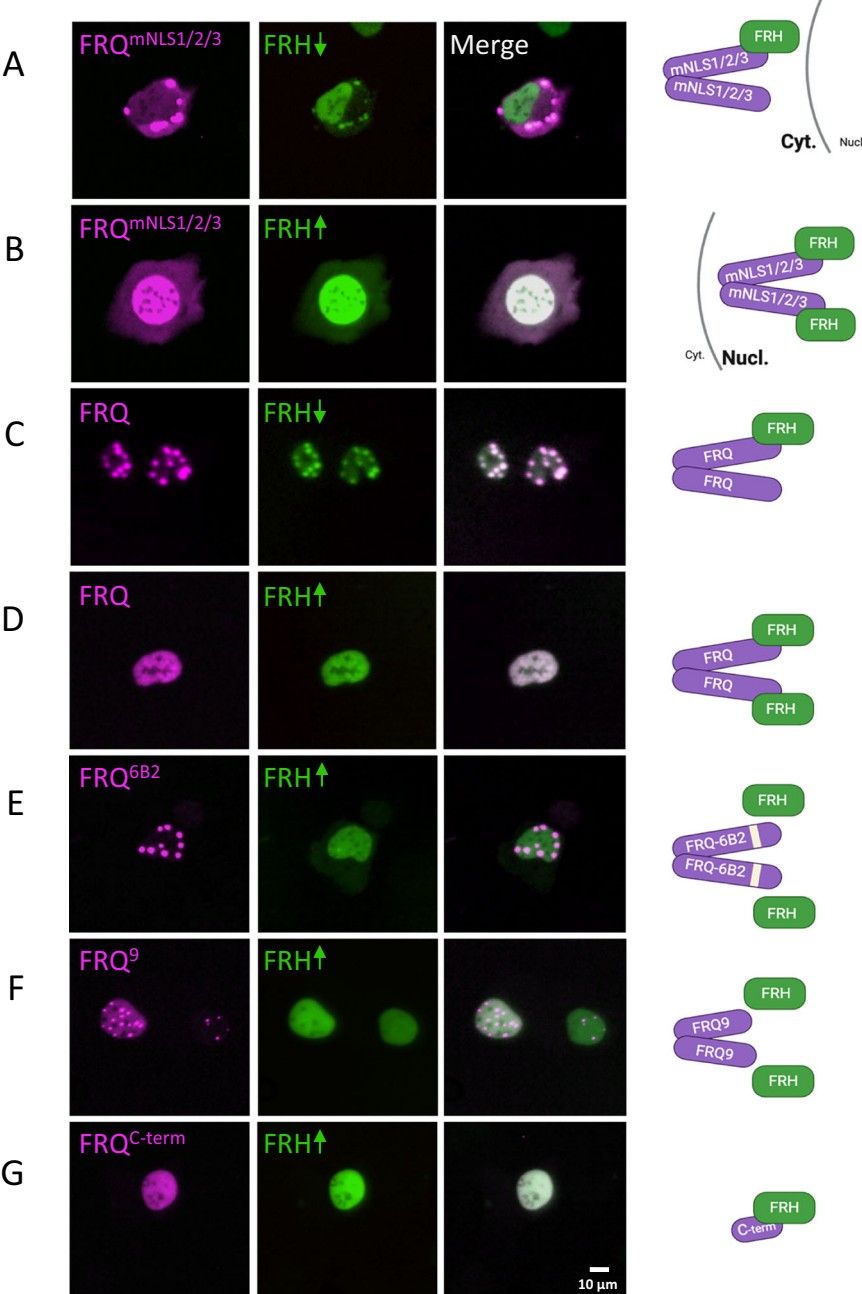

**Fig. 3 | Binding of saturating amounts of FRH to FRQ suppresses the formation of FRQ nuclear foci in U2OStx cells.** The indicated mK2-tagged FRQ versions were co-expressed with FRH-mNG. Interactions are represented schematically. **A** Co-expression of mK2-FRQ^mNLS1/2/3 (50 ng plasmid DNA) and non-saturating amounts of FRH-mNG (16.5 ng). mK2-FRQ^mNLS1/2/3 and FRH-mNG co-localize in cytoplasmic foci. **B** Co-expression of mK2-FRQ^mNLS1/2/3 (50 ng plasmid DNA) and saturating amounts of FRH-mNG (50 ng). mK2-FRQ^mNLS1/2/3 and FRH-mNG are dispersed in the nucleus. **C** Co-expression of mK2-FRQ and non-saturating amounts of FRH-mNG. mK2-FRQ and FRH-mNG co-localize in nuclear foci. **D** mK2-FRQ and saturating amounts of FRH-mNG. mK2-FRQ and FRH-mNG are dispersed in the nucleus. **E, F** 50 ng mK2-FRQ^6B2 or mK2-FRQ^9 with saturating amounts FRH-mNG. mK2-FRQ^6B2 and mK2-FRQ^9 form nuclear foci that do not contain FRH-mNG. **G** mK2-FRQ^C-term (50 ng) and saturating amounts FRH-mNG. Both proteins are dispersed in the nucleus. Panels show representative subcellular expression patterns. Quantitative data are summarized in Supplementary Table 1.

with additional FRQ regions that would otherwise remain undetectable. Such multivalent FRQ-FRH interactions have recently been demonstrated in peptide array studies[34].

## FRQ interacts with the WCC in absence of FRH

In *Neurospora*, WC-1 is rapidly degraded without its stabilizing partner WC-2 and, hence, cannot be easily analyzed[30]. We tagged WC-1 with mNG and WC-2 with mK2 to study their expression and localization in U2OStx cells. WC-1-mNG was well expressed and localized predominantly in the cytosol, while mK2-WC-2 localized to the nucleus and formed foci (Supplementary Fig. 2B). When expressed together, WC-1-mNG was depleted from the cytosol and co-localized with mK2-WC2 in nuclear foci (Supplementary Fig. 2C), indicating interaction of the two WCC subunits.

In *Neurospora*, the FFC complex interacts weakly and dynamically with the WCC[7,11] and inactivates WCC through phosphorylation by CK1a[11].

To investigate whether FRH is necessary for this interaction, we co-expressed in U2OStx cells WC-1-mNG with mK2-FRQ in absence of

WC-2 and FRH. WC-1-mNG, which localized mainly to the cytosol when expressed by itself (Supplementary Fig. 2B), co-localized with mK2-FRQ in nuclear foci (Fig. 4A), demonstrating that WC-1 by itself interacted with FRQ (in »90% of cells). Upon co-expression with mK2-FRQ$^{6B2}$, which has a mutation in the FRH binding site, WC-1-mNG was also recruited into the nucleus, but a substantial fraction of WC-1-mNG remained cytosolic (Fig. 4B). The data suggest that WC-1-mNG interacted with mK2-FRQ$^{6B2}$ but potentially with reduced affinity. In contrast, WC-1-mNG did not co-localize with the C-terminally truncated mK2-FRQ$^{9}$, which formed nuclear foci (Fig. 4C).

Next, we expressed mK2-WC-2 together with FRQ-mNG (Supplementary Fig. 2D, left). Both proteins formed nuclear foci, which overlapped, suggesting that WC-2 on its own also interacts with FRQ.

To study the interaction of FRQ with the WCC, we co-expressed mK2-FRQ with WC-1-mNG and untagged WC-2 (Fig. 4G) as well as FRQ-mNG with untagged WC-1 and mK2-WC-2 (Supplementary Fig. 2D, right). As expected, mK2-FRQ as well as FRQ-mNG formed nuclear foci. WC-1-mNG was enriched in all mK2-FRQ foci (Fig. 4D) and all FRQ-mNG foci contained mK2-WC-2 (Supplementary Fig. 2D), suggesting interaction of FRQ with WCC in absence of FRH. Some WCC foci remained separate, likely due to the saturation of the FRQ foci with WCC, causing excess WCC to accumulate in distinct foci.

When WC-1-mNG and untagged WC-2 were co-expressed with mK2-FRQ$^{6B2}$, WC-1-mNG was enriched in the mK2-FRQ$^{6B2}$ foci (Fig. 4E), indicating that the 6B2 region is not required for the interaction of FRQ with WCC. A few WC-1-mNG (WCC) foci remained distinct, suggesting excess of WCC over mK2-FRQ$^{6B2}$. In contrast, when WC-1-mNG and untagged WC-2 were co-expressed with mK2-FRQ$^{9}$, WC-1-mNG was not enriched in mK2-FRQ$^{9}$ foci (Fig. 4F).

To determine whether these expression patterns are cell-type specific, we also expressed the tagged clock proteins in HEK293T cells (Supplementary Fig. 3A, B). The tagged proteins displayed the same localization and colocalization patterns in HEK293T cells as in U2OStx cells.

Recent deletion studies suggested that the N-terminal half of FRQ is required for its interaction with WCC in *Neurospora*[39]. However, as these deletions are predicted to disrupt FRQ dimerization (e.g., FRQ$^{\Delta149-193}$) and CK1a recruitment (e.g., FRQ$^{\Delta482-510}$), they may impact WCC interaction indirectly.

The overexpression and co-localization in U2OStx cells indicate that the C-terminal third of FRQ is necessary with both WC-1 and WC-2, even in the absence of FRH. Because the 6B2 mutation in FRQ may weaken but does not eliminate WCC binding, it suggests that the WCC binding region overlaps with, but is distinct from, the FRH binding site. This raises the question of how FRH modulates this interaction.

## FRH blocks WCC binding to FRQ
To directly analyze whether WCC interacts with the C-terminal potion of FRQ, WC-1-mNG was co-expressed with mK2-FRQ$^{C-term}$ (Fig. 5A, left panels). WC-1-mNG; which is mainly cytoplasmic on its own, was recruited into the nucleus and uniformly dispersed just like mK2-FRQ$^{C-term}$. These findings indicate that the C-terminal third of FRQ is sufficient for interaction with WC-1. Thus, WCC and FRH both bind to the C-terminal third of FRQ. When WC-1-mNG and mK2-FRQ$^{C-term}$ were coexpressed with untagged FRH, mK2-FRQ$^{C-term}$ displayed nuclear localization, while WC-1-mNG remained predominantly in the cytoplasm (Fig. 5A, right), just like in the absence of mK2-FRQ$^{C-term}$ (see Supplementary Fig. 2B). Similarly, when WC-1-mNG and mK2-FRQ were co-expressed, WC-1-mNG was recruited from the cytosol into the nucleus and colocalized with mK2-FRQ in nuclear foci (Fig. 5B, left). In presence of FRH, mK2-FRQ was homogeneously dispersed in the nucleus, indicating its interaction with saturating amounts of FRH, while WC-1-mNG remained predominantly cytoplasmic (Fig. 5B, right).

To further analyze the impact of FRH on the interaction of FRQ with the WCC, we co-expressed WC-1-mNG and untagged WC-2

together with mK2-FRQ and either low amounts of untagged FRH, allowing co-localization of FRQ and FRH in nuclear foci, or with high amounts of FRH, interfering with foci formation (see Supplementary Fig. 4A, B). At low levels of FRH (Supplementary Fig. 4A, left), mK2-FRQ accumulated in nuclear foci. WC-1-mNG co-localized with these foci, suggesting WCC interacted with FRQ. At high levels of FRH (Supplementary Fig. 4A, right), mK2-FRQ was homogeneously dispersed in the nucleus, indicating that saturating amounts of FRH interfered with the self-interaction of mK2-FRQ. WC-1-mNG accumulated in nuclear foci, indicating that it assembled with untagged WC-2. The WC-1-mNG (WCC) foci, however, did not contain mK2-FRQ (Supplementary Fig. 4A, right). The data suggest that FRQ saturated with FRH does not interact with WCC.

For additional analysis, we expressed mK2-WC-2 and untagged WC-1 together with untagged FRQ and either high or low amounts of FRH-mNG. At low amounts, FRH-mNG was enriched in mK2-WC-2 (WCC) foci (Supplementary Fig. 4B, left), indicating that unlabeled FRQ in complex with sub-saturating amounts of FRH-mNG interacted with WCC. When expressed at a high level, FRH-mNG was dispersed in the nucleus, while mK2-WC-2 (WCC) foci persisted (Supplementary Fig. 4B, right).

The data strongly suggest that FRH is not only unnecessary for the interaction of FRQ with WC-1 and WC-2, but it may even interfere with it. WCC and FRH bind to overlapping but distinct sites within the C-terminal third of FRQ. The strong anchoring of FRH inhibits WCC binding when FRH is present in high, presumably saturating amounts, but permits WCC-FRQ interaction at lower, sub-saturating FRH concentrations. This finding was unexpected, as it had been suggested that FRH is necessary for the interaction between WCC and FFC[26]. This suggestion was based on data showing convincingly that WCC did not interact with FRQ complexes containing a mutant FRH with an R806H substitution.

To investigate further, we expressed WC-1-mNG and mK2-FRQ with either unlabeled FRH or FRH$^{R806H}$. Both FRH and FRH$^{R806H}$ competed with the nuclear recruitment of WC-1-mNG by mK2-FRQ (Supplementary Fig. 5). However, FRH$^{R806H}$ was a stronger competitor than FRH, supporting the notion that the R806H substitution may act as a dominant-negative mutation, leading to tighter binding to FRQ, potentially explaining why FRQ did not interact with WCC in the *frh$^{R806H}$ Neurospora* strain[26]. However, further analysis is required to uncover the mechanism underlying the R806H mutation.

## Phosphorylation of FRQ by CK1 induces its dissociation from FRH and nuclear export
Having shown that FRH blocks binding of WCC to FRQ, we asked whether and how the interaction of FRQ with WCC is regulated. In *Neurospora*, FRQ is gradually hyperphosphorylated by CK1a, which inactivates FRQ and favors its degradation[12–14]. To analyze the impact of CK1a, we co-expressed in U2OStx cells untagged CK1a with FRH-mNG and either mK2-FRQ, mK2-FRQ$^{6B2}$ or mK2-FRQ$^{9}$. The three FRQ proteins were hyperphosphorylated in a CK1a-dependent fashion (Fig. 6A). This indicates that CK1a phosphorylated mK2-FRQ in U2OStx cells while endogenous kinases did not support similar hyperphosphorylation of the overexpressed FRQ proteins.

Next, we examined the CK1a-mediated effect on mK2-FRQ localization by widefield microscopy 6 h after doxycycline (DOX) induction (Fig. 6B). In absence of CK1a, mK2-FRQ localized exclusively in nuclear foci (see also Fig. 1C), while mK2-FRQ formed nuclear and cytosolic foci when CK1a was co-expressed, indicating that the kinase promoted nuclear export of mK2-FRQ.

Subsequently, we examined the dynamics of CK1a-mediated effects on mK2-FRQ and FRH by Incucyte time-lapse live-cell imaging (Fig. 6C–E). Initially, mK2-FRQ and FRH-mNG co-localized in the nucleus. mK2-FRQ did not form foci, indicating that it interacted with saturation levels of FRH (Fig. 6C). After several hours, CK1a induced the

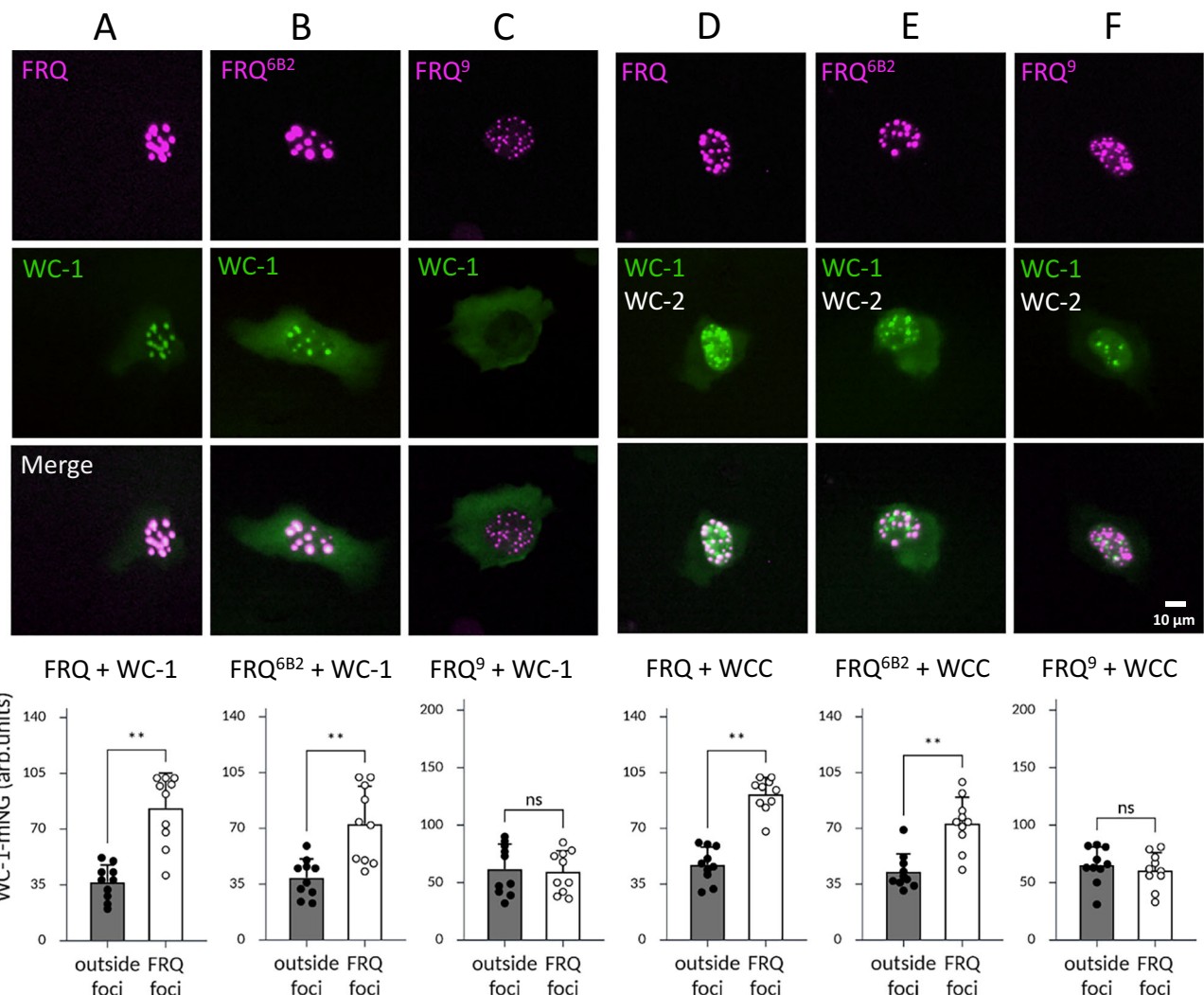

**Fig. 4 | FRQ interacts with WC-1 and WCC in absence of FRH. A–C** Co-expression of WC-1-mNG (*n* = 3). **A** with mK2-FRQ. WC-1-mNG is enriched in mK2-FRQ nuclear foci. **B** with mK2-FRQ^6B2. WC-1-mNG is enriched in mK2-FRQ^6B2 nuclear foci. **C** with mK2-FRQ^9. WC-1-mNG does not co-localize with mK2-FRQ^9 nuclear foci. **D–F** Co-expression of WC-1-mNG and untagged WC-2 (*n* = 3). **D** with FRQ. WC-1-mNG is enriched in mK2-FRQ nuclear foci. **E** with mK2-FRQ^6B2. WC-1-mNG is enriched in mK2-FRQ^6B2 nuclear foci. **F** with mK2-FRQ^9. WC-1-mNG is not enriched in FRQ^9-mNG nuclear foci. Lower panels: The enrichment of WC-1-mNG in foci of the indicated mK2-FRQ version over nuclear regions outside of foci was quantified. Foci from 10 cells were evaluated. Plotted is Mean +/− SD (exact Mean and SD can be found in Source Data file). One-tailed Wilcoxon signed-rank test: **A** and **E**: **\**p* = 0.006; **B** and **D**: **\**p* = 0.002; **C** and **F**: *p* = 0.3, ns not significant.

formation of large assemblies of mK2-FRQ in the cytosol, while FRH-mNG remained confined to the nucleus (Fig. 6C and Supplementary Fig. 6A). Interestingly, mK2-FRQ also formed focal assemblies in the nuclei (Fig. 6B, C), suggesting that FRH-mNG, despite being present in excess, could no longer saturate phosphorylated mK2-FRQ. In absence of CK1a, mK2-FRQ alone or together with FRH-mNG remained nuclear at all times and no cell with cytoplasmic FRQ foci was observed (Supplementary Fig. 6B). When CRM-1-dependent nuclear export was inhibited with leptomycin B, mK2-FRQ remained in the nucleus in all cells despite the presence of CK1a (Fig. 6C lower panels, Supplementary Fig. 6A). These data indicate that phosphorylation by CK1a caused dissociation of mK2-FRQ from FRH-mNG and supported its export from the nucleus. mK2-FRQ^6B2, which cannot interact with FRH-mNG, was also exported from the nucleus in an LMB-sensitive manner when co-expressed with FRH-mNG and CK1a, but with substantially faster kinetics than mK2-FRQ (Fig. 6D and Supplementary Fig. 6C), indicating that FRH bound to wt FRQ delayed its nuclear export. The data suggest that phosphorylation inactivated the positively charged NLSs, potentially through neutralization by negatively charged phosphosites or -clusters generated by CK1a. In contrast, mK2-FRQ^9 was not exported

from the nucleus when co-expressed with CK1a and FRH-mNG (Fig. 6E and Supplementary Fig. 6D), suggesting that the C-terminal third of FRQ contains a NES.

To narrow down the NES, we divided the C-terminal third of FRQ into three parts, deleting amino acid residues 631–756, 757–888, and 889–989 in mK2-FRQ (Supplementary Fig. 7A), respectively, and expressed each modified protein along with CK1a in U2OStx cells (Supplementary Fig. 7B). After DOX induction, we observed cytosolic accumulation of mK2-FRQ^{Δ631–756} and mK2-FRQ^{Δ889–989}, suggesting that CK1a-mediated phosphorylation favored cytoplasmic localization by inactivating the NLSs and/or activating the NES. In contrast, mK2-FRQ^{Δ757–888} remained nuclear, demonstrating its NES had been compromised by the deletion. Since this deleted segment also includes the FRH binding region, it seems likely that FRH can mask the NES, thereby regulating FRQ's nuclear export in a phosphorylation-dependent manner.

Our data indicate that in the absence of phosphorylation, the three NLSs in full-length FRQ dominate functionally over the NES, resulting in strong nuclear enrichment of FRQ. When FRQ is hyperphosphorylated by CK1a, it is exported to the cytosol, likely due to NLS

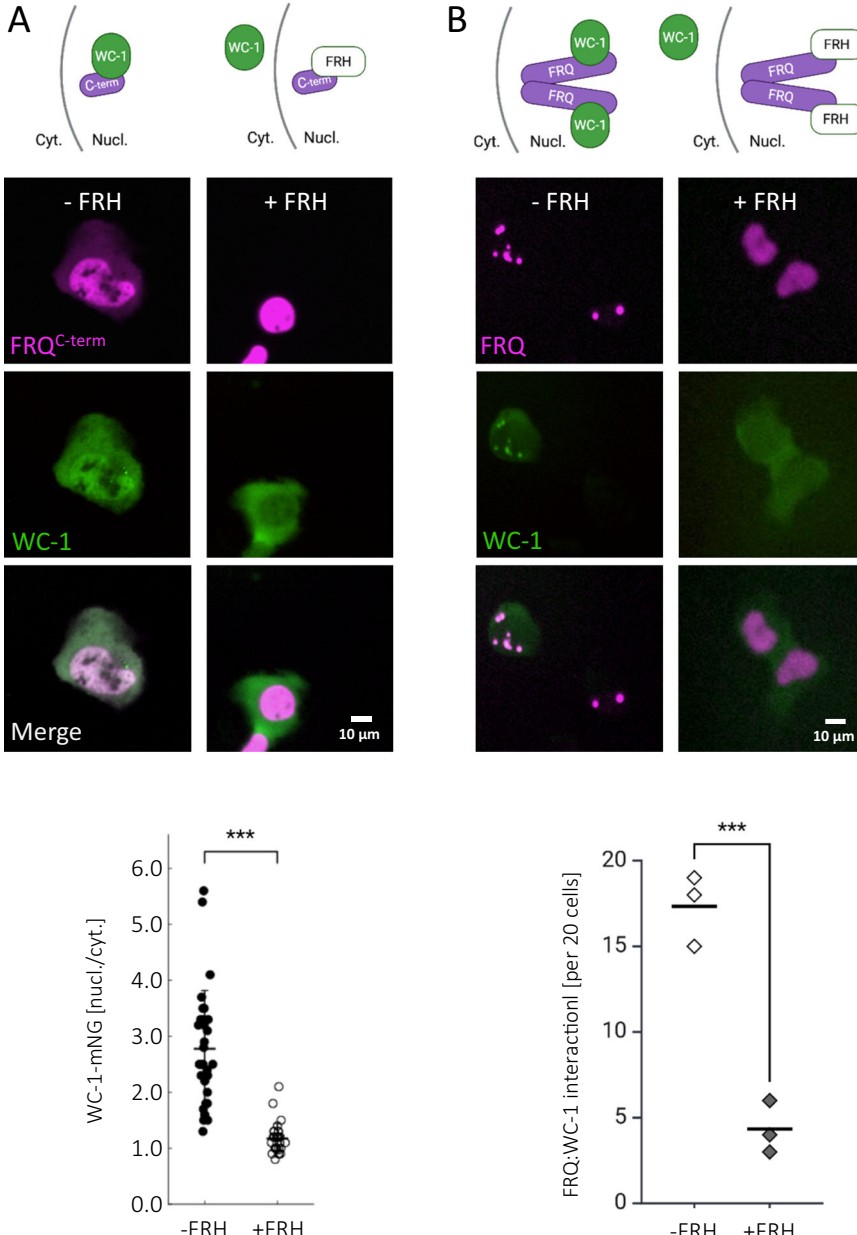

**Fig. 5 | FRH interferes with binding of FRQ to WC-1.** Schematics depict the interactions of fluorescently tagged (green and magenta) and untagged (white) proteins. **A** Co-expression of mK2-FRQ$^{C-term}$ and WC-1-mNG without FRH and with FRH. Left: In absence of FRH (left) mK2-FRQ$^{C-term}$ recruits WC-1-mNG into the nucleus. Right: FRH interferes with the recruitment of WC-1-mNG into the nucleus. The nuclear/cytoplasmic ratio of 30 cells per condition were analyzed with ImageJ. Plotted is Mean with SD "−FRH": 2.78+/− 1,04; "+FRH": 1.17+/− 0,27. ***: one-tailed

paired *T*-test, $p = 1.11 \times 10^{-8}$. **B** Co-expression of mK2-FRQ and WC-1-mNG without FRH and with FRH. Interaction of mK2-FRQ with WC-1-mNG was evaluated by counting cells displaying co-localization in nuclear foci (−FRH) or cells displaying nuclear enrichment of WC-1-mNG (+FRH), which is significantly reduced in presence of FRH. 20 cells each from three independent experiments were evaluated. ***: two-tailed Unpaired *T*-test, $p = 0.000328$, Mean is plotted: "−FRH": 16.80; "+FRH": 5.60.

neutralization. FRH slows this phosphorylation-driven export of FRQ by providing additional nuclear import capacity via its NLS and by masking FRQ's NES.

### Native FFC dissociates upon phosphorylation of FRQ by CK1a

In *Neurospora*, all detectable FRQ is present in a complex with FRH, which protects FRQ from degradation. FRQ$^{6B2}$ or FRQ$^9$, which cannot interact with FRH, are rapidly degraded. Hence, they accumulate only at low levels, despite being synthesized in excessive amounts due to nonfunctional negative feedback[29,40].

Since free FRQ does not accumulate at significant levels in *Neurospora*, it is challenging to investigate whether gradual hyperphosphorylation triggers its dissociation from FRH. Proteasome inhibitors are not very effective and lead to pleiotropic effects when administered at high doses and for longer time periods[41]. To circumvent degradation of free FRQ, we prepared native *Neurospora* extract from cells expressing 2 × FLAG-tagged FRH and hyperphosphorylated FRQ in vitro by adding recombinant CK1a and ATP (Fig. 7A). An untreated extract was used for comparison. Subsequently, 2 × FLAG-FRH was subjected to immunoprecipitation. In the control reaction without

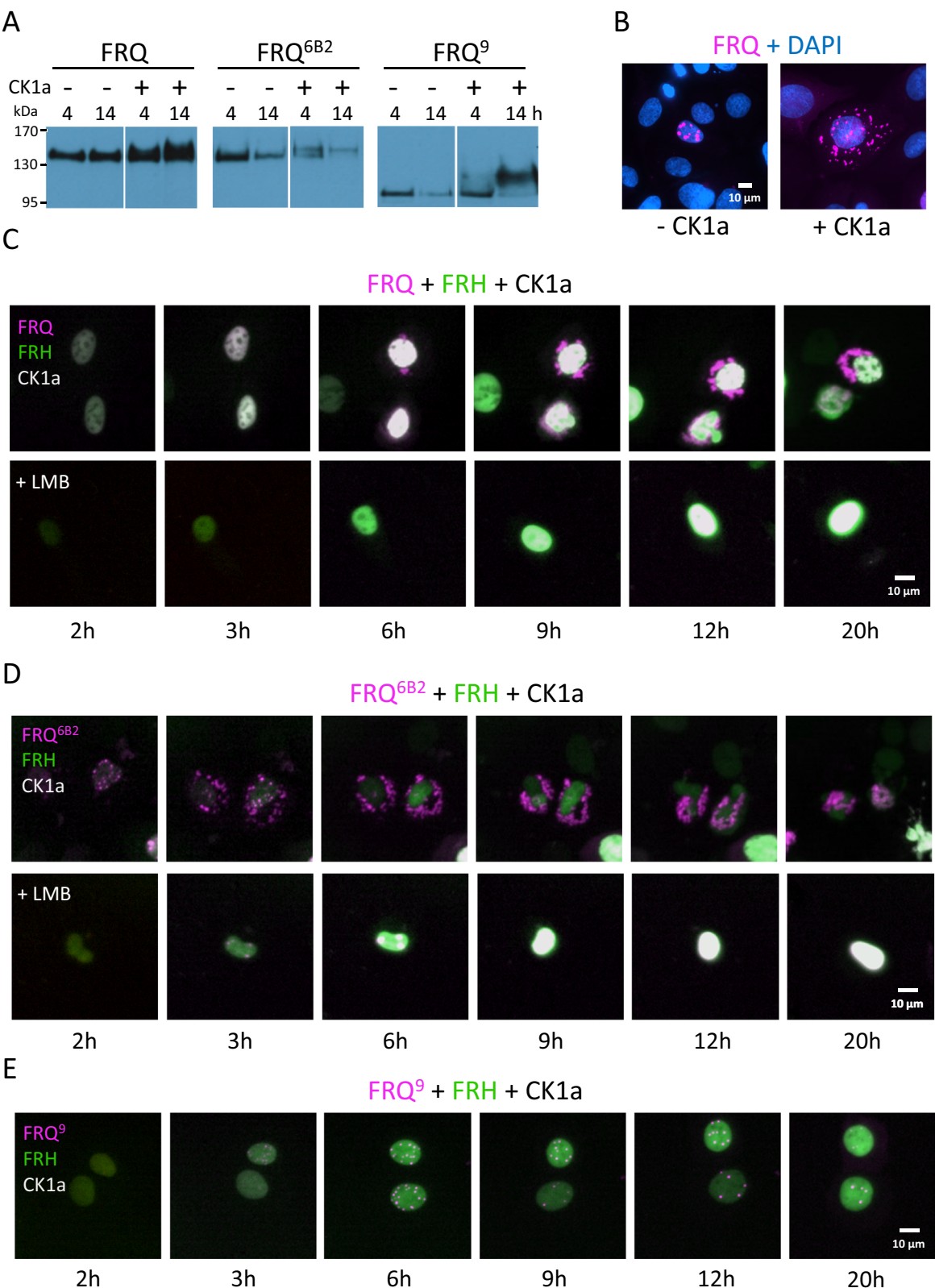

**Fig. 6 | Phosphorylation of FRQ by CK1a triggers its dissociation from FRH and nuclear export. A** Western blot analysis of mK2-FRQ, mK2-FRQ^6B2 and mK2-FRQ^9 co-expressed with and without CK1a in U2OStx cells for the indicated time periods. The samples were loaded side by side on the same gel in the order shown. They were cropped into separate panels in pairs to facilitate comparison. **B** Widefield fluorescence microscopy with DAPI staining of cells expressing mK2-FRQ with and without untagged CK1a. U2OStx were analyzed 6 h post DOX induction (*n* = 2). **C**–**E** Incucyte time courses of transfected U2OStx cells upon co-expression of FRH-mNG and untagged CK1a with **C** mK2-FRQ, **D** mK2-FRQ^6B2, and **E** mK2-FRQ^9. LMB was added where indicated (*n* = 3; LMB: *n* = 2).

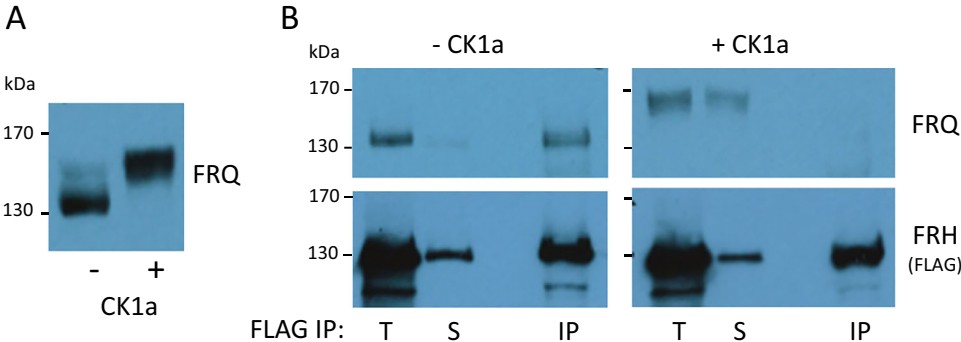

**Fig. 7 | Phosphorylation of native FRQ by CK1a triggers its dissociation from FRH. A** Whole cell lysate (WCL) of *Neurospora* expressing 2 × FLAG-tagged FRH prepared from cultures containing hypophosphorylated FRQ (grown 3 h in light after 10 h darkness) was incubated with recombinant CK1a and ATP, or left untreated. **B** Lysates from (**A**) were subjected to FLAG immunoprecipitation. Total (T), supernatant (S), and immunoprecipitate (IP) were analyzed by Western blot with antibodies against FRQ (upper panels and 2 × FLAG-FRH (Lower panels). Results shown are a representative of three independent experiments with reproducible results.

CK1a, hypophosphorylated FRQ co-immunoprecipitated with 2 × FLAG-FRH (Fig. 7B left). In contrast, upon incubation with CK1a and ATP, hyperphosphorylated FRQ no longer co-immunoprecipitated with 2 × FLAG-FRH (Fig. 7B right).

Size exclusion chromatography of untreated native *Neurospora* cell extract revealed that FRH co-eluted with FRQ in a high molecular mass fraction, indicating their interaction (Supplementary Fig. 8A, B). Given FRH's abundance relative to FRQ, most FRH molecules eluted in a lower molecular mass fraction, consistent with prior findings[36]. Conversely, when the cell extract was treated with CK1a and ATP, FRH no longer co-eluted with FRQ in the high molecular mass fraction (Supplementary Fig. 8A, B).

These findings strongly suggest that the phosphorylation of native FRQ by CK1a triggers its dissociation from FRH.

### Stoichiometry of FRQ-FRH complexes
FRQ is a dimer, and in *Neurospora* most FRQ dimers appear to bind a single FRH molecule[5], although monomeric FRQ variants are capable of binding FRH[36]. Our data from U2OStx cells suggest that FRQ dimers (FRQ$_2$) can assemble into heterotetramers with two FRH molecules (FRQ$_2$FRH$_2$). These tetramers can be converted into heterotrimers (FRQ$_2$FRH$_1$) by displacement of one FRH through CK1a-dependent phosphorylation of FRQ. Hence, in *Neurospora*, where CK1a is present, tetramers, trimers, and even FRQ dimers may accumulate in distinct proportions, depending on the circadian phase.

To directly assess the subunit composition of FRQ-FRH complexes in the absence of CK1a, we expressed the relevant proteins in HEK293T cells. FLAG-FRH, mK2-FRH, and mK2-FRQ were co-expressed, and native extracts were subjected to FLAG immunoprecipitation. mK2-FRH efficiently co-precipitated with FLAG-FRH (Fig. 8A, right panels). In contrast, in control IPs lacking either FLAG-FRH or mK2-FRQ, mK2-FRH remained in the supernatant (Fig. 8A, left and middle panels). These results demonstrate that dimeric FRQ can recruit two FRH molecules. Notably, a substantial fraction of mK2-FRH underwent nonspecific proteolytic clipping (see Fig. 8A left panel), generating an FRH fragment (FRH*) that co-migrated with FLAG-FRH and displayed similar behavior to full-length mK2-FRH.

To examine the subunit composition of FRQ-FRH complexes in *Neurospora*, we generated a strain expressing GFP-tagged FRH in addition to endogenous FRH. GFP pulldown revealed co-precipitation of very low levels of endogenous FRH (Fig. 8B). However, this was expected since FRH expression levels are roughly 10-fold higher than average FRQ levels[7,36] and FRH-GFP is expressed at levels similar to endogenous FRH. Under these conditions, only ~5% of endogenous FRH and ~5% of FRH-GFP molecules could be maximally bound to FRQ.

Therefore, even in the best-case scenario—where every FRQ dimer binds two FRH molecules—a maximum of only ~2.5% of endogenous FRH could be detected in a GFP pulldown. Given that FRH is progressively released through FRQ phosphorylation, the actual fraction of tetrameric complexes is likely substantially lower.

To verify that the low co-IP of endogenous FRH with FRH-GFP was genuine, we included two controls (Fig. 8C). First, a GFP pulldown from a strain lacking GFP-tagged FRH produced no endogenous FRH signal. Second, in a strain lacking FRQ (*frq¹⁰*), GFP-tagged FRH did not co-immunoprecipitate with endogenous FRH, confirming that the co-IP was mediated by FRQ.

Together, these findings strongly support that FRQ dimers can bind two FRH molecules in *Neurospora*, which are released stepwise upon CK1a phosphorylation.

Assuming that FRQ were produced at a constant rate, and the release of individual FRH molecules occurs independently, the steady-state ratio of FRQ$_2$FRH$_2$ to FRQ$_2$FRH$_1$ would be 1:2. Under physiological conditions, however, negative feedback suppresses FRQ synthesis, resulting in progressive conversion of FRQ$_2$FRH$_2$ into FRQ$_2$FRH$_1$. Consequently, FRQ$_2$FRH$_1$ would become the predominant FRQ species[5], while unbound FRQ$_2$ is degraded and does not accumulate.

## Discussion
We have previously shown that CK1a, anchored to FRQ, slowly hyperphosphorylates FRQ in a temperature-independent manner, forming a molecular module suitable for time measurement on a circadian scale[18]. However, the molecular mechanism underlying phosphorylation-dependent circadian timekeeping remains unclear. In this study, we provide data demonstrating that FRH, the folded binding partner of the intrinsically disordered FRQ, senses the time-dependent phosphorylation state of FRQ, forming an elaborate molecular timing device. The key features of this timer rely on a series of competitive interactions regulated by slow phosphorylation of FRQ by bound CK1a. We demonstrate that the slow gradual phosphorylation of FRQ initiates a two-step subunit remodeling process, first activating and then inactivation the FRQ-FRH complex in a time-dependent manner. Our data identify FRH as a decoding hub for temporal information encoded on FRQ through accumulated phosphate groups.

### FRH anchored to the C-terminal third of FRQ interacts with additional parts of FRQ
Using live-cell microscopy, we visualized the interactions and dynamics of *Neurospora* clock proteins in U2OStx cells. This heterologous

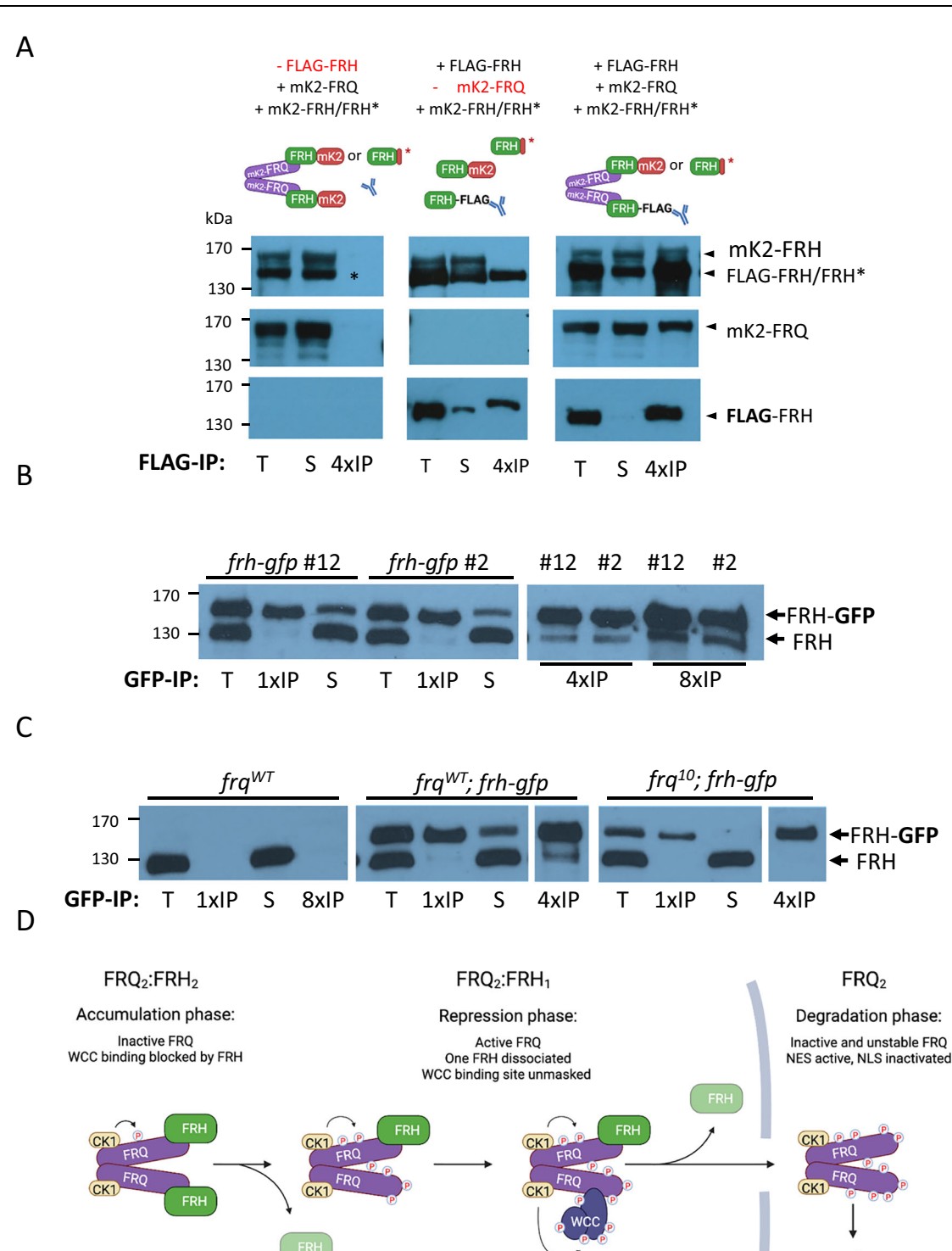

cell-based approach allowed us not only to characterize clock proteins but also to analyze their interactions and subcellular dynamics, which could not be examined in *Neurospora* or in a cell-free system with purified components. In U2OStx cells, overexpressed FRQ accumulated in nuclear foci, suggesting a propensity for self-interaction, which will be further characterized elsewhere. Interestingly, co-expression of high levels of FRH prevented the formation of nuclear FRQ foci but did not prevent foci formation by FRQ[9] (aa 1–662) or FRQ[6B2], which

both lack the FRH anchor site. These data suggest that anchoring FRH to the 6B2 region of FRQ increases the local concentration of FRH, favoring its weaker interaction with other regions of FRQ over FRQ self-interactions. In *Neurospora*, FRQ is expressed at lower levels but may also accumulate in larger assemblies[4]. It remains to be determined whether the self-interaction of FRQ and its suppression by FRH occur only at the level of a FRQ dimer or also in higher oligomeric assemblies.

**Fig. 8 | Subunit composition of the FRQ-FRH complex. A** FLAG-IPs from HEK293T lysates expressing the proteins schematically indicated in the top panels. Left panels: Control IP from cells expressing mK2-FRH and mK2-FRQ but no FLAG-FRH. Middle panels: Control IP from cells expressing mK2-FRH and FLAG-FRH but no mK2-FRQ. Right panels: FLAG IP from cells expressing mK2-FRH, mK2-FRQ and FLAG-FRH. *: In a substantial fraction of mK2-FRH the mK2-moiety was clipped off, giving rise to an untagged FRH moiety that comigrates with FLAG-FRH. 25 μg total lysate (T) and supernatant (S) and aliquots of the immunoprecipitates corresponding 100 μg (4 × IP) were analyzed by Western blot with antibodies against FRH (upper panel), FRQ (middle panel) and FLAG (lower panel) (n = 3). FLAG visualization was performed on a different gel using the same samples. **B** GFP pulldown from *Neurospora* lysates expressing FRH-GFP in addition to endogenous FRH. *frh-gfp* #2 and #12 represent two independent strains. T, 400 μg total lysate; S, aliquots of supernatant; IP, aliquots of immunoprecipitates corresponding to 1×, 4×, and 8× of the total lysate lane. Representative results from three independent experiments with reproducible results are shown. **C** GFP pulldown from *frq^{WT}* and from *frq^{WT}*, *frh-gfp* and *frq^{10}*, *frh-gfp* strains. Representative results from three independent experiments with reproducible results are shown. **D** Model of CK1a-dependent subunit remodeling in the FRQ-FRH complex. Newly synthesized, unphosphorylated FRQ accumulates with FRH in the nucleus as a heterotetramer with an $FRQ_2FRH_2$ stoichiometry. The nuclear accumulation of FRQ is facilitated by

its three nuclear localization signals (NLSs) and is further supported by a tight association with FRH, which provides a nuclear import signal and additionally mask FRQ's nuclear export signal (NES). The two bound FRH molecules block the WCC binding sites on FRQ, rendering the $FRQ_2FRH_2$ complex inactive. Consequently, the WCC remains active, promoting *frq* transcription and supporting the accumulation of high levels of inactive $FRQ_2FRH_2$. Phosphorylation of FRQ inactivates its NLSs and, after a considerable delay, triggers the dissociation of FRH. Due to the slow and partially stochastic nature of FRQ phosphorylation, the release of the two FRH molecules from a FRQ dimer is not synchronous, leading to the transient formation of a heterotrimeric $FRQ_2FRH_1$ species. The $FRQ_2FRH_1$ complex is the active species because it exposes a WCC binding site, allowing CK1a to phosphorylate and inactivate transiently interacting WCCs, thereby shutting down *frq* transcription. The $FRQ_2FRH_1$ complex can be exported via the exposed NES in one FRQ subunit but will be reimported via the FRH bound to the second FRQ subunit. Further phosphorylation of FRQ by CK1a eventually triggers the dissociation of the second FRH molecule, resulting in an $FRQ_2$-only complex. At this stage, FRQ's NLSs are fully inactivated by phosphorylation, prompting relocalization of $FRQ_2$ to the cytosol, where it can no longer support phosphorylation of nuclear WCC and is eventually degraded, thus closing the negative feedback loop. Subsequent dephosphorylation of WCC initiates a new circadian cycle. Created with BioRender.com.

## FRH dissociates from phosphorylated FRQ

Following a time delay, phosphorylation of FRQ by CK1a triggered the dissociation of the FRQ-FRH complex, resulting in the nuclear export of FRQ, while FRH remained in the nucleus of U2OStx cells. This was unexpected, as unbound FRQ is inherently unstable and rapidly degraded in *Neurospora*, which has hindered detailed analyses of its subcellular dynamics in absence of FRH. We further validated this phosphorylation-dependent dissociation by analyzing native FRQ-FRH complexes from *Neurospora* through co-immunoprecipitation and by size-exclusion chromatography.

## FRQ subcellular localization is regulated by three NLSs and one NES and by FRH

We found that FRQ contains three NLSs in the N-terminal two-thirds and one CRM1-dependent NES in the C-terminal third overlapping the FRH binding site. The NLSs are functionally dominant over the NES, leading to the nuclear accumulation of FRQ, including $FRQ^{6B2}$, which cannot bind FRH. This nuclear accumulation is further enhanced by FRH, which has its own NLS and masks the NES when bound to FRQ. Phosphorylation of FRQ by CK1a promotes its nuclear export, potentially by inhibiting the polybasic NLSs through interactions with phosphorylated regions in FRQ and by facilitating the release of bound FRH.

## FRH masks the WCC binding-site and is released upon phosphorylation of FRQ

It has been reported that specific mutations in the C-terminal third of FRQ affect its interaction with WCC but not with FRH[39]. We show here that FRQ interacts with both WCC subunits, WC-1 and with WC-2. The C-terminal third of FRQ is necessary and sufficient for the interaction with WCC. We found that the associations of FRQ with WCC and FRH are mutually exclusive. The binding sites in FRQ are overlapping but not identical, and the tight anchoring of FRH to unphosphorylated FRQ blocks the WCC binding site. This site becomes accessible only when FRH is released through FRQ phosphorylation by CK1a.

Our data suggest fundamental modifications of the *Neurospora* circadian clock model. Specifically, we show that the phosphorylation state of the intrinsically disordered FRQ dimer is decoded by its folded partner, FRH. Slow multisite phosphorylation of FRQ by CK1a triggers with a delay remodeling of an initially inactive nuclear heterotetrameric complex with $FRQ_2FRH_2$ stoichiometry (and bound CK1a) into an active trimeric $FRQ_2FRH_1$ complex that interacts with and inhibits WCC. Phosphorylation also inactivates FRQ's NLSs. The

trimeric $FRQ_2FRH_1$ complex may shuttle between compartments, exported via the NES exposed by release of FRH from one FRQ subunit and reimported as long as the six NLSs of the FRQ dimer are not fully inactivated by phosphorylation and in addition via FRH bound to the second FRQ subunit. Because FRQ phosphorylation is inherently slow and at least partially stochastic, the two FRH molecules are not released simultaneously. Instead, the second FRH is released only after a delay, leading to the formation of the $FRQ_2$ complex. In the nucleus, the $FRQ_2$ complex is active in negative feedback regulation. However, once exported to the cytosol, phosphorylated $FRQ_2$ cannot be reimported and is eventually degraded. This stepwise subunit remodeling of the FFC converts the steadily increasing phosphorylation state of FRQ into two distinct molecular switches: a delayed 'on-switch' that activates previously accumulated FRQ by exposing its WCC binding site, and an "off-switch" that inactivates FRQ by promoting its nuclear export and degradation (Fig. 8D).

Our model not only aligns with previous findings but also offers mechanistic insights into mutant phenotypes. For instance, our data suggest that the FRH^{R806H} mutant protein might block the WC-1 binding site more effectively than wild-type FRH. This increased binding affinity could explain the absence of WCC-FRQ interaction reported in *Neurospora* strains expressing FRH^{R806H}[26]. Additionally, monomeric mutants of FRQ, while able to interact with FRH and CK1, remain nonfunctional[6,36,42]. According to our model, monomeric FRQ in complex with FRH stays inactive due to its inability to bind WCC. After the phosphorylation-dependent release of FRH, monomeric FRQ exposes its WCC binding site and should be active in negative feedback. However, in the absence of a bound FRH, it will be rapidly exported and degraded in the cytosol, similar to $FRQ^{6B2}$, preventing it from accumulating in sufficient quantities to exert negative feedback in the nucleus. The mutations in the N-terminal region of FRQ, previously reported to impair WCC binding[39], fall within the predicted dimerization domain and therefore are more likely to affect FRQ dimerization than its direct interaction with WCC.

Several aspects of our model require further validation and refinement in future work. One question is whether the physiologically relevant form of FRQ in *Neurospora* is a dimer or a higher-order oligomeric assembly, and how these forms may be regulated. The nuclear-cytoplasmic distribution of FRQ appears to be influenced by the number and strength of its NLSs relative to NES. While we have demonstrated that an FRQ monomer contains three NLSs and one NES regulated by phosphorylation, further investigation is required to elucidate the precise mechanism by which

phosphorylation inactivates the NLSs. Because the polybasic NLSs are located near multiple phosphorylation sites, their gradual inactivation may result from charge neutralization. It is also essential to explore how phosphorylation of FRQ weakens the interaction with FRH to expose FRQ's NES and the WCC binding site. Perhaps the most compelling and exciting direction for future research, facilitated by the cell-based system established here, is to investigate how FRQ and CK1a influence the subcellular dynamics and chromatin association of the WCC.

While our model will undoubtedly evolve with future discoveries, it provides a detailed conceptual framework for understanding how this eukaryotic circadian clock measures time at the molecular level and offers an additional experimental system for further research in this field.

## Methods

### Plasmids
The N-terminal mKATE-2 (mK2) tag, derived from pmKate2-N and the C-terminal mNeonGreen (mNG) tag, derived from pMaCTag-07[43], were inserted into pcDNA™4/TO downstream of an inhibitory TetR-responsive cytomegalovirus (CMV-TetO) promoter by cloning with overlapping regions using appropriate primers[44]. In short, a linear vector and an insert DNA fragment carrying a 30 bp sequence overlap were transfected and recombined in vivo in *E. coli*. Accordingly, the cDNAs of the *Neurospora crassa* genes, *frh*, *ck1a*, *frq* and fragments of *frq* were inserted into pcDNA™4/TO, pcDNA™4/TO-mK2 and pcDNA™4/TO-mNG as indicated. The 6B2 mutation as well as the premature stop codon of the FRQ[9] mutant were introduced in the pcDNA™4/TO-mK2-*frq* by site directed mutagenesis using the Quik-Change II kit (Agilent).

FRQ-NLS mutants for expression in *Neurospora* were introduced by site directed mutagenesis into pBM60-*frq*, resulting in pBM60-*frq*$^{mNLS1}$, *frq*$^{mNLS2}$, −*frq*$^{mNLS3}$ and −*frq*$^{mNLS1/2/3}$ (QuikChange II kit, Agilent). For FRH-GFP, pMF309[31] (expressing β-tub-gfp under the control of the ccg-1 promoter) was used as source vector for cloning and the *tub* ORF replaced by *frh*.

Primers and Oligos used for cloning are listed in Supplementary Data 1.

### Western blots
Western blotting was performed as previously described[45]. Chemiluminescence signals were detected using X-ray films, which were then developed in Konica Minolta SRX-101A Medical Film Processor. Ponnçeau staining of Western blots was used as loading control (Supplementary Data).

### U2OS T-REx and HEK293T cells and culture conditions
U2OS T-REx (U2OStx) and HEK293T (ATCC) cells were maintained in a 5% $CO_2$ incubator at 37 °C in Dulbecco's modified Eagle's medium (DMEM, Thermo Fisher Scientific) supplemented with 10% fetal bovine serum (Thermo Fisher Scientific), 1% penicillin-streptomycin (Thermo Fisher Scientific), and Hygromycin B (50 μg/mL, Invivogen)

Leptomycin B (Merck; CAT# L2913-.5UG) was added at a final concentration of 20 nM one hour after DOX induction.

### Protein expression
U2OStx cells were seeded in 96-well plates 24 h prior to transfection (Corning, CAT#3598). A fixed amount of plasmid DNA (150 or 200 ng as indicated), pcDNA™4/TO plasmids with the genes of interest balanced with empty vector, was transfected per well using Xfect™ Transfection Reagent (Takara Bio) according to the manufacturer's instructions. Protein expression was induced 24 h post transfection by addition of 10 ng/mL DOX. pcDNA4/TO-FRQ-mNG was co-transfected with pcDNA4/TO-FRQ at a ratio of 25 ng: 50 ng to avoid overexposure of mNG fluorescence.

HEK293T cells (60–80% confluence) were transfected using an in-house PEI reagent (adapted from Longo et al., [46]). For transfection, 15 μg plasmid DNA was mixed with PEI at a 4:1 ratio in 2 mL DMEM. After 24 h, cells were scraped (Sarstedt, Cat. #83.3950) and lysed in cell lysis buffer (25 mM HEPES pH 7.4, 140 mM NaCl, 2 mM EDTA, 0.5% Triton ×-100, 1 mM NaF) supplemented with PMSF (1 mM), leupeptin (5 μg/mL), and pepstatin A (5 μg/mL). Protein extracts were sonicated for 5 min in an ultrasonic bath (Merck) and then collected after centrifugation. Protein concentration was measured using a NanoDrop (PeqLab).

### Protein extraction for Western blot
Cells were seeded in 12-well plates 24 h before transfection. At either 4 or 14 h after DOX induction, cells were harvested directly in 2× protein sample buffer and boiled at 95 °C for 7 min before Western blot analysis.

### IncuCyte live-cell microscopy
After induction of transfected U2OStx cells with DOX, 96-well plates were placed in either the Incucyte ZOOM® or Incucyte® SX1 system (Sartorius) at 37 °C. Images were acquired at 20× magnification at 1-h intervals for 48 h and then analyzed using the Incucyte® 2023 A GUI (Sartorius) and Incucyte® 2016B GUI (Sartorius) software.

### Widefield microscopy
U2OStx cells were transfected and induced with DOX. Six hours post-induction, cells were fixed with 4% PFA for 15 min at room temperature, followed by multiple washes with PBS. Coverslips were mounted using ProLong™ Glass Antifade Mountant with NucBlue™ Stain (Cat. #P36981) and sealed with nail polish. Imaging was performed on a Ni-E widefield microscope (Nikon Imaging Center, Heidelberg) using the DAPI (EX 390/18, DM 416, EM 460/60) and Texas Red (EX 562/40, DM 593, EM 624/40) filter sets.

### Quantification and statistical analysis
IncuCyte analysis: In each experiment, 20–30 cells were evaluated. Cells with fluorescence levels that were either too low or excessively high were excluded from the analysis. The cytoplasmic and nuclear distributions of mNG- and mK2-tagged proteins and their co-localization were examined as specified in each respective experiment. Interaction of FRQ and WC-1 or FRQ and WCC (Fig. 4) was determined by co-localization of WC-1-mNG in mK2-FRQ nuclear foci. ImageJ was used to analyze fluorescence intensity in ten cells. Regions of interest (ROIs) of $3 × 3$ pixels were selected within nuclear foci formed by mK2-FRQ ("FRQ foci") and within nuclear regions lacking foci ("outside foci"). The mean gray value of the green channel (WC-1-mNG) was measured for each ROI. The resulting values were plotted using BioRender (Created in BioRender.com).

### Nuclear/cytoplasmic distribution of WC-1-mNG
The nuclear/cytoplasmic ratio of WC-1-mNG in Celess expressing mK2-FRQ$^{C-term}$ +/− FRH was determined using thresholding in ImageJ, as described by the Keith R. Porter Imaging Facility (University of Maryland, https://kpif.umbc.edu/image-processing-resources/imagej-fiji/determining-fluorescence-intensity-and-area). Briefly, to label the nucleus, the thresholding function was applied to the red channel (mK2-FRQ$^{C-term}$). This selection was then applied to the green channel (WC-1-mNG), where the threshold function was used to define the entire cell, thus allowing differentiation between the nucleus and the cytosol. The nuclear/cytoplasmic ratio was calculated by comparing the mean gray values of the nuclear and cytoplasmic regions.

### *Neurospora crassa* culture conditions
Conidial suspensions in 1 M sorbitol were prepared from strains grown (5–7 days) on standard solid growth medium (2.2% agar, 0.3% glucose,

**Table 1 | *Neurospora crassa* strains used in this study**

| Strain | Genotype | Relevant phenotype | Source |
|---|---|---|---|
| *wt* | ras-1[bd] | frq[WT] | |
| Δfrq; frq[WT], frq-lucPEST | ras-1[bd], Δfrq::HygB[R], his3::frqWT tub2::pfrq-lucPest-ttrpC | frq[WT] with frq-lucPEST reporter | Cesbron et al.[47] |
| Δfrq; frq[mNLS1], frq-lucPEST | ras-1[bd], Δfrq::HygB[R], his3::frq[mNLS1] tub2::pfrq-lucPest-ttrpC | frq[mNLS1] with frq-lucPEST reporter | This work |
| Δfrq; frq[mNLS2], frq-lucPEST | ras-1[bd], Δfrq::HygB[R], his3::frq[mNLS2] tub2::pfrq-lucPest-ttrpC | frq[mNLS2] with frq-lucPEST reporter | This work |
| Δfrq; frq[mNLS3], frq-lucPEST | ras-1[bd], Δfrq::HygB[R], his3::frq[mNLS3] tub2::pfrq-lucPest-ttrpC | frq[mNLS3] with frq-lucPEST reporter | This work |
| Δfrq; frq[mNLS1/2/3], frq-lucPEST | ras-1[bd], Δfrq::HygB[R], his3::frq[mNLS1/2/3] tub2::pfrq-lucPest-ttrpC | frq[mNLS1/2/3] with frq-lucPEST reporter | This work |
| frh-gfp | ras-1[bd], his3::pccg1-frh-gfp | expression of additional FRH C-terminally tagged with GFP | This work |
| frq[10]; frh-gfp | ras-1[bd], frqKO::HygB[R], his3::pccg1-frh-gfp | expression of additional FRH C-terminally tagged with GFP in a frq deficient strain | This work |

0.17% L-arginine, 1× Vogel's medium, and 0.1% biotin). Standard growth medium for liquid cultures contained 2% glucose, 0.17% L-arginine, and 1× Vogel's medium. Liquid cultures were inoculated with conidia and grown in constant light at 25 °C for 2 to 3 days if not indicated otherwise.

To express hypophosphorylated FRQ, 48 h light-grown liquid cultures were transferred to the dark for 10 h (essentially leading to the degradation of "old" FRQ) and then exposed to light for 3 h to induce a burst of FRQ expression before mycelia were harvested and native protein extract was prepared. Strains used in this study are listed in Table 1.

**Native protein extraction.** To extract native protein from *Neurospora*, mycelial tissue was ground in liquid nitrogen using a precooled mortar and pestle. The resulting powder was suspended in an extraction buffer containing 50 mM Hepes-KOH (pH 7.4), 137 mM NaCl, 10% (v/v) glycerol, and 5 mM EDTA, with 1 mM phenylmethylsulfonyl fluoride (PMSF), leupeptin (5 μg/ml), and pepstatin A (5 μg/ml). Protein concentration was measured using a NanoDrop (PeqLab). Subcellular fractionation was performed as described ref. 37. 100 mg of total extract and cytosol and 50 mg of nuclei (determined by Bradford assay) were loaded on an SDS PAGE.

**Luciferase reporter assay.** The luciferase assay was performed as described previously[47] with slight modifications: 96-well plates containing growth medium including 75 μM D-luciferin were inoculated with $3 \times 10^4$ conidia per well and incubated in constant darkness at 25 °C for 3 days. The plate was then transferred to an incubator at 25 °C with an EnSpire Multilabel Reader (Perkin Elmer). Bioluminescence was measured in 30 min intervals. The following light/dark regime was used: 2 h darkness, 12 h light, 12 h darkness, 12 h light, constant darkness. Light intensity used was 30 μE. Data was normalized to average bioluminescence levels during light exposure.

**Signal processing and damped sinewave analysis of luciferase data.** Signal data were processed and analyzed using a custom Python script with AI assistance of Google Gemini, which can be provided upon request. The analysis focused on the time series data starting at $t = 40.0$ h (=shortly after cultures were released into constant darkness, i.e., into free-run). Raw signal data were first smoothed to reduce noise by applying a 1D uniform filter with a window size of ±4 h. This smoothed signal was then detrended by subtracting a linear regression line to remove any long-term drift or baseline shift. This process isolated the oscillatory component of the signal for subsequent fitting. The damping coefficient (γ) was determined by fitting the detrended data to a damped sinewave model using a least-squares optimization method (scipy.optimize.curve fit). The model used is

given by the equation:

$$f(t) = A \cdot e^{-\gamma t} \cdot cos(\omega t + \phi) + C$$

where $A$ is the initial amplitude, $\gamma$ is the damping coefficient, $\omega$ is the angular frequency, $\phi$ is the phase shift, and $C$ is the vertical offset. Initial guesses for the parameters, particularly for the angular frequency ($\omega$), were informed by a Fast Fourier Transform of the detrended data to ensure a robust fit. The damping coefficient (γ) was calculated as an output parameter of this fitting process. The period ($\tau$) was calculated using the fundamental relationship:

$$\tau = 2\pi/\omega.$$

**In vitro phosphorylation.** For in vitro phosphorylation of FRQ, *Neurospora* whole cell lysate was incubated over night at 4 °C with 2 μg purified recombinant CK1a[18] per mg of lysate in a buffer containing a final concentration of 50 mM Hepes/KOH pH 7.4, 120 mM NaCl, 11.3 mM MgCl$_2$, 12.5 mM ATP, 1 × PhossStop (Roche), 1 mM PMSF, 1 μg/ml leupeptin, 1 μg/ml pepstatin.

**Co-immunoprecipitation.** FLAG Co-IP was performed as described[36]. Briefly, 20 μl of M2 FLAG Sepharose beads (Sigma) were washed twice in PBS. 4 mg of whole cell lysate from a strain expressing N-terminally 2 × FLAG-tagged FRH was then added to the beads and filled to 500 μl with PBS supplied with protease inhibitors (Roche) and incubated for 3 h at 4 °C. Supernatant was removed, beads washed twice, and bound protein eluted by boiling at 95 °C for 5 min in 2× Laemmli. 400 μg of the input and supernatant were loaded together with amounts of IP corresponding 1× (for FLAG/FRH decoration) and 9× (for FRQ decoration) equivalents of the input. Western blots were probed with antibodies against FRQ and FLAG (FRH).

GFP pulldown was performed with GFP-Trap agarose beads (Chromotek). 40 μl of bead slurry was equilibrated in PBS supplemented with protease (PI) and phosphatase (P-stop) inhibitors. For each reaction, 8 mg of protein extract in 200 μl extraction buffer was adjusted to a final volume of 600 μl with PBS + PI + P-stop and incubated with equilibrated beads at 4 °C for 6 h on a rotating wheel. Supernatant was removed, beads washed twice, and bound protein eluted by boiling at 95 °C for 5 min in 100 μl 2× Laemmli. For SDS−PAGE analysis, equivalents of 400 μg Total extracts (T) and supernatant (S) were loaded. For immunoprecipitation (IP), 1×, 4×, and 8× equivalents of 400 μg were analyzed.

**Size exclusion chromatography.** A Superose 6 Increase G10/300 GL column was used in an Äkta Pure system (GE Life Sciences/Cytiva). The column was equilibrated with gelfiltration buffer (25 mM HEPES/ KOH

(pH 7.4), 140 mM NaCl, 1 mM EDTA pH 8.0, 1% glycerol, 0.05% Triton ×-100). 10 mg native *Neurospora* protein extracts were loaded onto a 200 μL loop and chromatography was performed at 4 °C with a flow rate of 0.5 mL/min. Fractions of 490 μL were collected, and protein was precipitated by adding 122.5 μL of 50% TCA (w/v). The samples were incubated on ice for 10 min and then centrifuged (4 °C, 14,000 × *g*, 10 min). Pellets were washed with 1 mL acetone, air-dried for 3 min at room temperature, and resuspended in 90 μL of 2× Laemmli buffer. Samples were boiled at 95 °C for 5 min. 45 μL was loaded on SDS-PAGE, followed by Western blotting for analysis.

### Reporting summary

Further information on research design is available in the Nature Portfolio Reporting Summary linked to this article.

## Data availability

All data are available from the corresponding author upon request. Source data are provided with this paper.

## Code availability

The code used for the analysis of the data in this study is available from the corresponding author upon request.

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

## Acknowledgements

We thank Alessia Ruggieri and Michael Knop for providing mKATE-2 and mNeonGreen templates, respectively. We thank Fidel E. Serrano for help with widefield microscopy. The work was supported by the Deutsche Forschungsgemeinschaft, TRR186.

## Author contributions

C.S.: U2OStx Incucyte experiments and analysis, HEK293T cell experiments. B.R.: some U2OStx Incucyte experiments. S.S. & A.C.R.D.: *Neurospora* experiments and analysis. L.L.: GFP-FRH pulldowns. M.B.: Conceptualization, writing.

## Funding

## Competing interests

The authors declare no competing interests.
