## [Transparent Peer Review file · Nature Communications]

Casein Kinase 1a Mediates a Two-Step Subunit Remodeling Mechanism to Regulate the FRQ-FRH Circadian Clock Complex

Corresponding Author: Professor Michael Brunner

This file contains all reviewer reports in order by version, followed by all author rebuttals in order by version.
Redactions – published data

Version 1:

Reviewer comments:

Reviewer #1

(Remarks to the Author)

Schunke et al. developed a heterologous cell-based system (U2OS cells) to study Neurospora clock proteins using live-cell microscopy. They expressed FRQ, WCC, FRH, and CK1a, analyzing FRQ localization, FRQ-WCC/FRH interactions, and CK1a-mediated FRQ phosphorylation. Their findings suggest CK1a-dependent phosphorylation remodels the FRQ-FRH complex in a two-step process. While this system offers novel insights into clock regulation, the conclusions require validation in Neurospora, and the omission of key literature weakens the contextual framework. Addressing these points would strengthen the study's impact.

Major points:

1. Figure 1, “mK2-FRQ6B2 formed nuclear foci like full-length mK2-FRQ and mK2-FRQ9 formed nuclear foci”. This is not real in Neurospora, in which FRQ is mainly localized in the cytosol, despite its essential role in the nucleus (PMID: 9482720, PMID: 21300798). Although the authors also created these relevant strains in Neurospora, they do not examine their localization, nor cited prior localization studies. One explanation is that CK1a is not co-expressed with FRQ in the cells, as indicated by the data in Figure 6. However, it remains puzzling to discuss the localization of FRQ in U2OS cells when it does not align with the in vivo published data in Neurospora.
2. DAPI staining is essential to confirm nuclear localization. This is needed at least in Figure 1 or Figure S1, if not in other figures.
3. Figures 1 and 2, the authors found that three NLS are required for FRQ localization. Previous work identified NLS1 as critical for FRQ nuclear localization and clock function (PMID: 9482720). The authors report that only combined NLS1-3 mutations disrupt FRQ nuclear localization and reduce rhythm amplitude. Since they generated Neurospora NLS mutants (Figure 2), they should assess FRQ localization in these strains (NLS1, NLS2, NLS3, and NLS1-3).
4. Figure 6A and Figure S5A, “After several hours, CK1a induced the formation of morphous assemblies of mK2-FRQ 286 in the cytosol, while FRH-mNG remained confined to the nucleus (Fig. 6B, S5A)”. The claim that CK1a expression shifts FRQ to the cytosol is unsupported; Figure S5A shows FRQ remains nuclear. This contradicts expectations and requires clarification.
5. Figure S6B, the subcellular distribution of this mutant mK2-FRQ Δ 631-756 (nuclear vs. cytosolic) is unclear and should be explicitly addressed.
6. “Phosphorylation of FRQ by CK1 induces its dissociation from FRH and nuclear export”. This conclusion also contradicts previous work (PMID: 9482720, PMID: 21300798). Previous studies indicate that FRQ phosphorylation does not significantly influence its cellular localization but does affect its interaction with WCC. In this study, CK1a expression plays an important role in FRQ localization. Could they discuss the differences?
7. Figure 8, “Slow multisite phosphorylation of FRQ by CK1a triggers with a delay remodeling of an initially inactive nuclear hetero-tetrameric complex with FRQ2FRH2 stoichiometry (and bound CK1a) into an active trimeric FRQ2FRH1 complex that interacts with and inhibits WCC”. Is the ratio between FRQ and FRH derived from the overexpression of proteins in U2OS cells or from existing literature? If it is based on overexpression in U2OS cells, it may not be accurate for Neurospora. Without evidence from Neurospora or supporting literature, the model should either omit precise ratios or cite relevant studies.
8. The authors miss many relevant literatures. A part of them is listed below.
References (PMID: 38526948, PMID: 24316221) are required when the authors mentioned IDR of FRQ in “The FRQ protein,

85% of which consists of intrinsically disordered regions (IDRs)², dimerizes via a coiled-coil in its N-terminal region 3” or other relevant part.

The references (i.e. PMID: 31554810, PMID: 34182774) are required for “FCD-bound CK1a is sufficient to phosphorylate FRQ on a circadian timescale and in a temperature-insensitive manner, and thus the FFC constitutes a molecular module suited to measure time 16,18.”

References are needed here, “For example, CK1a and FRH have essential functions outside the clock and cannot be deleted or inactivated. FRQ and WC-1 without their binding partners, FRH and WC-2, respectively, are not expressed at a sufficient level to pursue complex biochemical analyses.”

References (PMID: 12098706, PMID: 12098705) are required here, “In addition, the WCC is a light receptor that resets the circadian clock in response to blue light stimuli.”

References are required for “In *Neurospora*, WC-1 is rapidly degraded without its stabilizing partner WC-2 and, hence, cannot be easily analyzed.”

Minor points :

1. Figure 3C-E, The FRH:FRQ ratio is noted in the legend but not the main text.
2. For Figures S1 and S5, Combine multi-panel figures onto single pages for clarity.
3. “It has been reported that specific mutations in the C-terminal third of FRQ affect its interaction with WCC but not with FRH”: “third” or “part” ?

Reviewer #2

(Remarks to the Author)

This manuscript addresses the question of the functional roles of FRQ phosphorylation by CK1a and FRH binding, two aspects of FRQ function that are clearly important for circadian rhythms in the fungal model system *Neurospora crassa*, but which lack a firmly established mechanistic role. There have been challenges studying CK1a and FRH in *Neurospora* because they are both essential genes. The authors established a heterologous expression system for the fungal proteins in human U2OS osteosarcoma cells, enabling them to look at how FRH binding and CK1a phosphorylation affect nuclear localization and interaction with white collar (WC) proteins, which serve as the central transcriptional regulator of the *Neurospora* circadian clock. They conclude that the well-documented slow hyperphosphorylation of FRQ by CK1a remodels the FRQ-FRH-CK1a (FFC) complex, and make the important discovery that FRH competes with the white collar complex (WCC) and regulates nuclear localization and protein turnover, providing a new model that helps to explain the biochemical basis for the delay in FFC regulation of the WCC in the clock mechanism. Their evidence for this new model is compelling, even if it mostly originates from the heterologous system. Mutations based off the U2OS data are tested in *Neurospora* cycling assays and some *in vitro* biochemistry provides strong support for the central claim of this work. Overall, this is an exciting piece of work that offers a new explanation for the delay in transcriptional regulation in a circadian clock, leading to many additional studies that can follow up in more detail on the specific sites and mechanisms. However, looking at the figures alone, it is often hard to synthesize the important points, significantly detracting from the impact of the work. There are some comments and suggestions below for improvements that might help convey this information better to readers.

1. It might be helpful to denote graphically which constructs in Figure 1 can interact with FRH. Also, the microscopy images could be smaller and perhaps be accompanied by a graphical summary of the data to assist in interpretation at a glance.
2. It would be nice to get damping rates for the data in Figure 2.
3. The use of red-green coloring throughout the figures should be changed in consideration of color-blind viewers.
4. It might help to have cartoons of the constructs and interactions alongside the microscopy images in Figure 3, 4, and 6, similar to Figure 5.
5. The data in Fig. S7 is compelling and complementary to the co-IP in Figure 7. I suggest including it in the main figure.
6. It would be nice to have data of FRH NLS mutant(s) to support the claim made in Figure 3e and lines 177-180.

Reviewer #3

(Remarks to the Author)

Summary:

This paper explores the molecular events governing FRQ localisation, participation in complexes and stability using combined mutational and microscopy-based approaches. Owing to the intractability of *N. crassa* to live cell microscopy the authors elect to establish a heterologous overexpression model (using U2OS cells) to explore the interplay between FRH, FRQ and phosphorylation (of FRQ) by casein kinase 1. They characterise a series of FRQ mutants in terms of their nuclear/cytoplasmic distribution and relate this to downstream effects on the circadian clock. They later use the same mutants to study both the distribution and colocalization of FRQ with various binding partners.

The findings regarding FRH’s regulatory influence on FRQ localization and interactions are of substantial interest to the wider circadian biology community. However, several methodological and interpretive blind-spots should be addressed before the manuscript is suitable for publication.

Strengths:

This manuscript provides a compelling and novel system for dissecting the spatial regulation and molecular interactions of *Neurospora crassa* clock proteins using a heterologous human cell model. Using this platform, the authors identify sites for

regulation of nuclear import and export and demonstrates the role of FRH in modulating localisation. They link the nuclear mis-localisation of different NLS mutants to circadian phenotypes in *N. crassa*. In general, the paper is clearly written and well presented, and it does strongly suggest a novel role for FRH in regulating the interaction of WCC and FRQ.

I am also interested if the authors can rationalise the different NLS mutation phenotypes (Figure 2), i.e. NLS2 and NLS3 have opposing effects on circadian period, which is an interesting and counter-intuitive (at least to me) observation. Further to this, does introducing the 6B2 mutation to native FRQ abolish rhythmicity, particularly when combined with the triple NLS mutant. Similarly, would a FRQ-NES mutant have a similar circadian phenotype.

Weaknesses:

1. In general, quantification and appropriate statistical tests should be provided for all microscopy data and western blots.
2. Studying typically low-abundance proteins in the context of over-expression has inherent problems, it is plausible that some of the interactions which are reported here are, in part, a function of their being present at a significantly higher copy number than would be expected in *N. crassa*, though this is caveated by the authors. In particular, the formation of nuclear aggregates is common consequence of over-expressing transcriptional factors, e.g., Buchberger et al., PLoS One, 2013. Given the size of the foci (several microns) it is plausible that FRQ and FRH may be in nuclear aggregates/inclusions or interacting with other large macromolecular assemblies, RNA, other membrane less organelles or nuclear bodies in a fashion that does not reflect their endogenous function and sub-nuclear localisation. It would be revealing to perform some simple IP or pull-down experiments with affinity-tagged constructs.

Furthermore, over-expression artefacts are possibly exacerbated by the combined choice of host (U2OS) and the highly processive CMV-tet promoter. Since the promoter is tet-responsive it should be possible to titrate the expression level with of at least one tagged protein with doxycycline. Furthermore, where plasmids have been co-transfected one cannot assume equal partitioning of transcriptional and translational machinery, i.e. the observed response when combining different mutants is not just a feature of the biochemical activities of the expressed proteins. Moreover, although fine at the population level it is not reasonable to assume that each dually fluorescent cell has taken up an equimolar ratio of plasmids to that in the transfection mixture unless validated e.g. by calibrating the relative fluorescence intensities. Since the authors are using a heterologous system, I would have greater confidence in the conclusions of this paper if some microscopy could be recapitulated in at least one other cell-line. This would demonstrate that what is observed here is due to the intrinsic properties of the specific proteins and not cell-line specific.

2. The authors make claims about the overlapping, or not, of WC1 and FRQ foci in Figure 4H-I, respectively. Firstly I do not agree that this is particularly apparent in the figures themselves, nonetheless one would expect some degree of overlap even if they were independently and randomly distributed. Frq-9 signal also appears somewhat more diffuse than Frq-6B2 and so one may make the interpretation it is simply interacting with WC at a lower stoichiometry. Proper quantification of the degree of colocalization should be provided using established methods.

The method of quantification for figure 5A is unusual and rather opaque. It appears that cells are classified as exhibiting either nuclear or cytoplasmic localisation of WC-1-mNG, based on the relative fluorescent intensities from each compartment. This approach fails to capture the spectrum of localization, from predominantly nuclear to predominantly cytoplasmic. Nonetheless, since observations are not independent (as one goes up the other goes down), neither are they continuous, therefore it invalidates the choice of statistical test. I would recommend a simple nuclear/cytoplasm ratio be reported instead, this would preserve underlying biological variation and better facilitate comparison between +FRH/-FRH conditions.

3. The authors make multiple allusions to the ability of FRQ to homodimerize and form the basis of various intermediate complexes. Though reduced foci formation by smaller fragments is consistent with a role of the entire molecule forming intermolecular multivalent interactions it doesn't necessarily prove that foci formation is caused by a biologically relevant assembly process (phase-separation or otherwise), nor does it necessarily mean they are functionally important for clock output. Some FRAP experiments on the various FRQ foci would go some way to demonstrating this. Regarding figure 6, the blot lacks quantification and loading control, without this one cannot claim that endogenous CK1 or other kinases are not phosphorylating FRQ. To aid the comparison and draw a parallel with the blot, it would be helpful to have some live cell microscopy without CK1a overexpression.
4. Though the model the authors propose (Figure 8) is certainly plausible, it is not clear to me what is the direct evidence for the existence of the specific intermediates in the data presented (i.e. gradual dissociation of FRH), which is surprising given the suggested slow phosphorylation kinetics. Additional experiments (i.e. Native-PAGE/crosslinking etc.) would be required to interrogate the relative stoichiometries of the complex participants.

Minor point: The Y-axes of Figure 2 are not consistent ('.' and ','). It would be helpful to add the amplitude and damping rates if they intend to comment on these in the text.

Version 2:

Reviewer comments:

Reviewer #1

(Remarks to the Author)

The authors have addressed my concerns.

Reviewer #2

(Remarks to the Author)

The authors have done an acceptable job addressing my prior comments. The addition of the cartoons makes it easier to move between the text and figures to process the information. There are still many questions remaining that will require future studies to dissect, but given that this type of biochemical study is not possible in the fungal system, I think there is value in proposing this new model. Although the conclusions here depend on overexpression in a heterologous system, I am satisfied with the biochemical data supporting the conclusions.

I do have two minor comments that should be easy to address.

The new co-IP data in figure 8 should state in the figure what protein is being precipitated. This is done in figure 7 and it is standard practice.

The cartoon present in figure 8 has different coloring than the ones used throughout the other figures. It may make sense to unify the coloring here.

Reviewer #3

(Remarks to the Author)

In the revised manuscript the authors have made a comprehensive attempt to address several important points. However, some methodological and interpretative concerns remain about the steps the authors have taken to mitigate the risk of confirmation bias, because standard practises such as loading controls, quantification, statistical analyses are still not consistently employed throughout.

More generally the authors are of course entitled to their opinions, and presumably find their interpretation very convincing. Whilst I find their interpretations plausible, I do not personally find the supporting evidence even comes close to being unambiguous. If the authors are reluctant to do further experiments and analyses then I would encourage them to tone down the confidence with which they present their interpretation. Starting with the abstract, I suggest that they should be clear to distinguish their interesting experimental observations from their informed speculations.

More detailed responses are below:

Original point 1. In general, quantification and appropriate statistical tests should be provided for all microscopy data and western blots.

Answer:

Western blots show qualitative data:

Reviewer response:

Despite the qualitative nature of the data, loading controls are still recommended to ensure equal loading/transfer and not addressed in the methods. Tubulin loading controls are shown for supplementary figure 1. It is not clear why they are not shown for other blots.

Microscopic images:

Fig 1: The vast majority (if not all) cells expressing the indicated FRQ variants exhibit the subcellular localization shown in the corresponding figure panels.

Fig 3

Same answer, the vast majority (if not all) cells expressing the indicated FRQ variants exhibit the subcellular localization shown in the corresponding figure panels. Specifically, we have evaluated 20 cells per condition.

Fig 4

We have evaluated 20 cells per condition.

A-E: interaction of FRQ with WC-1:

Fig. A,B: 20/20 cells exhibit the subcellular localization shown in the corresponding figure panels.

Fig. 4C: 19/20 cells exhibit the subcellular localization shown in the figure panel.

Fig. 4D is quantified in Fig 5A.

E: 18/20 cells exhibit the subcellular localization shown in the figure panel.

F-I: interaction of FRQ with WCC:

Fig. F-G: 20/20 cells exhibit the subcellular localization shown in the figure panels.

I: 19/20 cells exhibit the subcellular localization shown in the figure panel. i.e. no co-localization of WCC with FRQ9.

Reviewer response:

Using established metrics of co-localization would add rigor to this study and add substance to the claims made in the text. If 20 cells were evaluated then why not show the quantification of the evaluation.

Original point 2. Studying typically low-abundance proteins in the context of over-expression has inherent problems, it is plausible that some of the interactions which are reported here are, in part, a function of their being present at a significantly higher copy number than would be expected in *N. crassa*, though this is caveated by the authors.

In particular, the formation of nuclear aggregates is common consequence of over-expressing transcriptional factors, e.g., Buchberger et al., PLoS One, 2013. Given the size of the foci (several microns) it is plausible that FRQ and FRH may be in nuclear aggregates/inclusions or interacting with other large macromolecular assemblies, RNA, other membrane less organelles or nuclear bodies in a fashion that does not reflect their endogenous function and sub-nuclear localisation. It would be revealing to perform some simple IP or pull-down experiments with affinity-tagged constructs.

Furthermore, where plasmids have been co-transfected one cannot assume equal partitioning of transcriptional and translational machinery, i.e. the observed response when combining different mutants is not just a feature of the biochemical activities of the expressed proteins. Moreover, although fine at the population level it is not reasonable to assume that each dually fluorescent cell has taken up an equimolar ratio of plasmids to that in the transfection mixture unless validated e.g. by calibrating the relative fluorescence intensities.

Answer:

All clock protein interactions in *Neurospora* were previously identified through pulldown assays of wild-type and mutant proteins conducted by us and others. These interactions are validated/reproduced in our co-localization assays, with specificity confirmed by the absence of co-localization in the respective mutants. Moreover, the co-immunoprecipitation of FRH with FRQ in HEK293T cells (Fig. 8A) provides additional evidence that the co-localizing proteins physically interact.

Suitable ratios of co-transfected plasmids were determined empirically. In U2OS cells, individual cells express varying amounts of the proteins. We monitored the cells over time and analyzed only those that expressed both tagged proteins at medium levels (i.e., excluding cells with extremely low or extremely high expression of one or both proteins).

The reviewer is of course correct. Detectable interaction depends on ratios, but also on concentrations and binding affinities. For example, when analyzing the interaction of WC-1 with FRQ, we always see colocalization of WC-1 in FRQ nuclear foci but observe in several cases WC-1 remaining in the cytosol, likely due to WC-1 being in excess over FRQ or expression levels not permitting saturation binding to nuclear FRQ. Our primary focus, however, is on whether interaction occurs - e.g., colocalization of WC-1 with nuclear FRQ foci - or whether there is no interaction, such as no colocalization of WC-1 with FRQ9 foci.

Reviewer response: Although evidence from *Neurospora* is supportive of the author's general thesis, for the previously mentioned reasons, foci formation in of itself does not provide sufficient proof that localization occurs through the specific interactions mentioned. To establish U2OS overexpression as a useful system for studying mechanisms of *Neurospora* protein complexes there is an obligation to demonstrate that foci are not artefactual. The IP presented in figure 8 is a step towards this but it is unclear why this was done only in HEK293 and not also in U2OS.

Moreover, it is very well to say that cells are selected based on a medium level of expression, but it is not clear from the text how this is determined.

Original point 2. The authors make claims about the overlapping, or not, of WC1 and FRQ foci in Figure 4H-I, respectively. Firstly I do not agree that this is particularly apparent in the figures themselves, nonetheless one would expect some degree of overlap even if they were independently and randomly distributed. Frq-9 signal also appears somewhat more diffuse than Frq-6B2 and so one may make the interpretation it is simply interacting with WC at a lower stoichiometry.

Answer:

The C-terminally truncated FRQ9 protein(G663TER) does not interact with WC-1 in *Neurospora*. Truncations beyond aa663 are WCC binding deficient (e.g. Wang et al. 2023 PMID 37220856), which is consistent with our observations, i.e. WC-1 never colocalized with FRQ9 foci (comp. Figs. 4A,C):

Reviewer response: The point I intended to make is that Figure H and Figure I appear similar. How this interaction occurs in *Neurospora* is somewhat irrelevant. The onus is on the authors to demonstrate this system faithfully reflects endogenous interactions. Without proper quantification of the degree of localization, the reader is in the position of having to take the authors explanations of the images in good faith.

Original point 4. Though the model the authors propose (Figure 8) is certainly plausible, it is not clear to me what is the direct evidence for the existence of the specific intermediates in the data presented (i.e gradual dissociation of FRH), which is surprising given the suggested slow phosphorylation kinetics. Additional experiments (ie. Native-PAGE/crosslinking etc.) would be required to interrogate the relative stoichiometries of the complex participants.

Answer:

We show pulldown from HEK293T cells and *Neurospora*, demonstrating that a FRQ dimer can bind two FRH molecules (see answer to all reviewers above).

Concerning the stepwise release, and assuming that the CK1 α -dependent dissociation of the two FRH molecules from the FRQ dimer is not strictly coupled but occurs independently, we provide here a theoretical calculation to illustrate the effect of

slow phosphorylation on FRH release:

Assume a FRQ dimer initially binds two FRH molecules. If slow phosphorylation by CK1 triggers FRH release at a rate of e.g. 0.25 h^{-1} ($t_{1/2} \approx 2.8 \text{ h}$, median), the average waiting time (mean) for dissociation of the first FRH - producing an FRQ_2FRH_1 species - equals $t = 1/(2 \times 0.25 \text{ h}^{-1}) = 2 \text{ h}$.

The second FRH dissociates after an additional average waiting time of $1/(0.25 \text{ h}^{-1}) = 4 \text{ h}$. Consequently, in this theoretical example, the average time delay before inactive FRQ_2FRH_2 becomes active via release of the first FRH is 2 h, and the resulting time window during which species with only one FRH bound (FRQ_2FRH_1) remain active is 4 h.

If FRQ were produced at a steady rate, the ratio of FRQ_2FRH_2 to FRQ_2FRH_1 would be about 1:2. In a physiological context (Neurospora), where negative feedback operates to shut down FRQ synthesis, FRQ_2FRH_2 would be converted to FRQ_2FRH_1 , which would become the predominant FRQ species (FRQ₂ is degraded and does not accumulate). The experimental observation that FRQ_2FRH_2 are detected in HEK293T cells and with the expected low abundance in Neurospora (see new Fig. 8) is consistent with the stepwise release model.

Reviewer response/Figure 8:

The figure legend and main text appear to suggest different experimental set-ups.

Main text:

“FLAG-FRH and mK2-FRH were co-expressed with, or without mK2-FRQ”

Figure legend:

“FLAG-IP from HEK293T lysates containing FLAG-FRH and mK2-FRH with and without FLAG-FRH”

In any case, I found this figure and the accompanying text confusing. The way the blots are presented is not helpful. Some of the blots are overloaded, and it is not clear what the arrows are pointing to in the upper panel of 8a.

Critically, this experiment is not suitable to infer complex stoichiometry, SDS-PAGE is a bulk measurement and cannot definitively establish the presence of said complexes.

8b/c similarly to 8a is not suitable to infer stoichiometry. The combined experiments only weakly support the suggested model.

“To directly examine the subunit composition of FRQ-FRH complexes in the absence of CK1a, we expressed the proteins in HEK293T cells”. This statement is not strictly true, HEK293T cells also possess endogenous CK1a.

Version 3:

Reviewer comments:

Reviewer #3

(Remarks to the Author)

Thank you for trying to accommodate my feedback

RESPONSE to REVIEWER COMMENTS

To all Reviewers:

Dear Reviewers,

in our revised manuscript, we added a new paragraph at the end of the Results section to directly analyze the subunit composition of the FRQ-FRH complex in both HEK293T cells and *Neurospora* (Fig. 8).

To this end, we performed a pulldown of FLAG-tagged FRH from HEK293T cells co-expressing mK2-FRH with or without FRQ. In the presence of FRQ, we observed co-immunoprecipitation of mK2-FRH with FLAG-FRH, demonstrating that a FRQ dimer can simultaneously bind two FRH molecules.

We also generated a *Neurospora* strain expressing, in addition to endogenous FRH, a GFP-tagged FRH copy. A GFP pulldown revealed co-precipitation of endogenous FRH. **IMPORTANTLY**, interpretation of this result requires consideration of the relative expression levels:

FRH is expressed at roughly **10-fold** higher levels than FRQ, and FRH-GFP is expressed at levels comparable to endogenous FRH. Thus, when immunoprecipitating FRH-GFP, at most ~5% of the FRH-GFP molecules could be bound to FRQ. In the maximal scenario where every FRQ dimer binds two FRH molecules, half of the FRQ molecules in the pulldown would carry two FRH-GFP, and the other half would carry one FRH-GFP plus one endogenous FRH. Consequently, **only ~2.5% of endogenous FRH would be detectable** in the assay. Moreover, because the heterotetrameric FRQ₂FRH₂ complex is continuously converted into the trimeric FRQ₂FRH₁ complex, and negative feedback eventually suppresses FRQ synthesis, only a weak endogenous FRH signal is expected under physiological conditions.

To confirm that the co-IP signal is genuine, we performed two **controls**. First, a GFP pulldown from a strain lacking GFP-tagged FRH produced no endogenous FRH signal. Second, in an *frq*¹⁰ strain (FRQ deletion), GFP-tagged FRH did not co-precipitate endogenous FRH, indicating that co-IP is mediated by FRQ dimers. Together, these findings strongly support that FRQ can bind two FRH molecules in *Neurospora*.

Hurley et al. (2013; PMID: 24316221), using a strain expressing FLAG-tagged and V5-tagged FRH, reported FRQ bound to a single FRH as the predominant species. However, the sensitivity of their assay likely precluded detection of less abundant FRQ dimers containing both a V5- and a FLAG-tagged FRH.

Reviewer #1 (Remarks to the Author):

Schunke et al. developed a heterologous cell-based system (U2OS cells) to study *Neurospora* clock proteins using live-cell microscopy. They expressed FRQ, WCC, FRH, and CK1a, analyzing FRQ localization, FRQ-WCC/FRH interactions, and CK1a-mediated FRQ phosphorylation. Their findings suggest CK1a-dependent phosphorylation remodels the FRQ-FRH complex in a two-step process. While this system offers novel insights into clock regulation, the conclusions require validation in *Neurospora*, and the omission of key literature weakens the contextual framework. Addressing these points would strengthen the study's impact.

Major points:

1. Figure 1, "mK2-FRQ6B2 formed nuclear foci like full-length mK2-FRQ and mK2-FRQ9

formed nuclear foci". This is not real in *Neurospora*, in which **FRQ is mainly localized in the cytosol**, despite its essential role in the nucleus (PMID: 9482720, PMID: 21300798).

Answer:

It has been shown that in *Neurospora* FRQ is concentrated in the nucleus, where it is active in negative feedback (see Fig's below).

However, the nuclei in filamentous fungi like *Neurospora* occupy only about 5-10 % of the cytosolic volume (I roughly calculated approximately 7% of the volume). Hence, the large fraction of cytosolic FRQ is in part due to the low nuclear/cytosolic volume ratio. In addition, our data suggest that the shuttling equilibrium of the FRQ2FRH1 heterotrimer, which is the major FRQ species in *Neurospora* (Hurley et al., 2013; PMID: 24316221) contributes to the cytosolic fraction.

.... Although the authors also created these relevant strains in *Neurospora*, they do not examine their localization, nor cited prior localization studies. One explanation is that CK1a is not co-expressed with FRQ in the cells, as indicated by the data in Figure 6. However, it remains puzzling to discuss the localization of FRQ in U2OS cells when it does not align with the in vivo published data in *Neurospora*.

Answer:

As suggested, we examined the localization of FRQ in *Neurospora*. All FRQ-NLS mutants, including FRQ^{NLSmut1/2/3}, display similar subcellular distribution like wt FRQ (Fig. S1D), as FRH binding alone is sufficient to direct FRQ to the nucleus. In U2OStx cells, FRQ^{NLSmut1/2/3} remains cytosolic in the absence of FRH but becomes nuclear in its presence (Fig. 3A,B). Our subcellular localization data from U2OS cells are consistent with published observations from *Neurospora* (see answer to query #3).

2. DAPI staining is essential to confirm nuclear localization. This is needed at least in Figure 1 or Figure S1, if not in other figures.

Answer:

Unfortunately, the Incucyte system does not have the capability to detect DAPI. Therefore, nuclei were identified in all experiments using phase-contrast imaging. See example:

Furthermore, by increasing the brightness/contrast of the images, low levels of proteins become detectable outside the foci. Unlike the foci, this fraction is uniformly distributed throughout the nucleus, providing a reliable internal nuclear marker as shown below for Fig. 3C:

In addition, free FRH is consistently localized to the nucleus and can be used to identify nuclei in the Incucyte instead of DAPI. Similarly, WC-2 is always nuclear.

Yet, as recommended by the reviewer, we include representative wide field microscopic images in which nuclei are visualized with DAPI staining (Fig. 1C and Fig. 6B).

3. Figures 1 and 2, the authors found that three NLS are required for FRQ localization. Previous work identified NLS1 as critical for FRQ nuclear localization and clock function (PMID: 9482720). The authors report that only combined NLS1-3 mutations disrupt FRQ nuclear localization and reduce rhythm amplitude. Since they generated *Neurospora* NLS mutants (Figure 2), they should assess FRQ localization in these strains (NLS1, NLS2, NLS3, and NLS1-3).

Answer:

As suggested, we show the localization of FRQ mutants in *Neurospora* (see response to 1.). The absence of nuclear localization of FRQ-ΔNLS (now NLS1) reported in PMID: 9482720 could not be reproduced. In our hands, FRQ lacking NLS1 remains nuclear in *Neurospora*. This fact has been recognized for a long time. (Görl, 2002, PhD thesis, p. 80). Notably, FRQ⁹, which lacks its NES, is almost exclusively nuclear (see below & Fig. S1D).

C = cytosol, N = nucleus

4. Figure 6A and Figure S5A, “After several hours, CK1a induced the formation of amorphous assemblies of mK2-FRQ in the cytosol, while FRH-mNG remained confined to the nucleus (Fig. 6B, S5A)”. The claim that CK1a expression shifts FRQ to the cytosol is unsupported; Figure S5A shows FRQ remains nuclear. This contradicts expectations and requires clarification.

Answer:

The reviewer is right, there are also FRQ foci in the nucleus, in addition to cytosolic FRQ. Our statement did not exclude this. Yet, we changed the sentence:

After several hours, CK1a induced the formation of amorphous assemblies of mK2-FRQ in the cytosol and in the nucleus, while FRH-mNG remained confined to the nucleus (Fig. 6B, S5A). This observation is fully consistent with our model and indeed matches our expectations. According to our model, FRQ is exclusively cytosolic only when all three NLSs are inactivated and both FRH molecules are released (FRH possesses its own NLS and can shuttle FRQ back into the nucleus). Because phosphorylation is rather stochastic, numerous intermediates exist in which, even after some time of CK1a phosphorylation, one or more NLSs remain active and/or one FRH remains bound. These intermediates distribute between the cytosol and nucleus, depending on their specific shuttling equilibrium. Figure S5A shows the individual channels for FRQ and FRH, demonstrating that CK1a-phosphorylated FRQ is present in both compartments, cytosol and nucleus.

We show in the revised version (Fig S5B) that without CK1a, FRQ as well as FRQ+FRH are consistently nuclear, demonstrating that CK1a is required for its export.

We also provide a widefield microscopic image with DAPI (Fig. 6B) showing CK1a-dependent cytoplasmic localization of mK2-FRQ.

Together, the data suggest that the wild-type FRQ remaining in the nucleus in the presence of CK1a still has FRH bound.

Notably, FRQ does not form nuclear foci when saturated with FRH, but does so at subsaturating FRH levels (see Fig. 3C, D). In Fig. 6C (saturating FRH conditions), FRQ fails to form foci after 3 h of incubation with CK1a. However, after prolonged incubation with CK1a (e.g. 12 h), FRQ forms foci in the nucleus (and cytoplasm) despite excess FRH (homogeneously dispersed) in the nuclei, indicating that at least one FRH molecule has dissociated by the action of CK1a.

5. Figure S6B, the subcellular distribution of this mutant mK2-FRQ Δ 631-756 (nuclear vs. cytosolic) is unclear and should be explicitly addressed.

Answer:

We have replaced the panel for mK2-FRQ Δ 631-756 with one showing clearer cytoplasmic localization. Below are three additional examples for the CK1a-induced cytosolic localization of mK2-FRQ Δ 631-756:

6. “Phosphorylation of FRQ by CK1 induces its dissociation from FRH and nuclear export”. This conclusion also contradicts previous work (PMID: 9482720, PMID: 21300798). Previous studies indicate that FRQ phosphorylation does not significantly influence its cellular localization but does affect its interaction with WCC. In this study, CK1a expression plays an important role in FRQ localization. Could they discuss the differences?

Answer:

This is precisely why we expressed the clock proteins heterologously in U2OStx cells, as some FRQ species are rapidly turned over and do not accumulate to significant levels in *Neurospora*.

Our data suggest that the FRQ dimer remains cytosolic only when both FRH molecules are dissociated by CK1a and sufficient NLSs (the FRQ dimer has 6 NLSs) are inactivated. In *Neurospora*, FRQ dimers lacking FRH are unstable, rapidly degraded, and therefore do not accumulate in the cytosol at substantial levels, making reliable biochemical analysis impossible. (e.g. below an experiment from our lab showing that FRQ does not accumulate in a *Neurospora* strain expressing the mutant FRH ^{Δ 100-150} (Hurley et al., 2013; PMID: 24316221) that cannot interact with FRQ.)

In *Neurospora*, most cytosolic FRQ species likely contain a single FRH molecule bound per FRQ dimer (Hurley et al., 2013; PMID: 24316221).

According to our model, unmodified FRQ dimers with two FRH molecules reside in the nucleus, while partially phosphorylated FRQ species with one FRH bound shuttle between the cytosol and nucleus. Given the high cytosol-to-nucleus volume ratio in filamentous fungi (~5%), these substoichiometric complexes are mainly cytoplasmic and may represent the predominant FRQ species detected in *Neurospora*. This explains why the majority of detectable FRQ in *Neurospora* is cytosolic and likely contains substoichiometric amounts of FRH (PMID: 24316221), despite each FRQ molecule having its own FRH binding site.

7. Figure 8, “Slow multisite phosphorylation of FRQ by CK1a triggers with a delay remodeling of an initially inactive nuclear hetero-tetrameric complex with FRQ2FRH2 stoichiometry (and bound CK1a) into an active trimeric FRQ2FRH1 complex that interacts with and inhibits WCC”. Is the ratio between FRQ and FRH derived from the overexpression of proteins in U2OS cells or from existing literature? If it is based on overexpression in U2OS cells, it may not be accurate for *Neurospora*. Without evidence from *Neurospora* or supporting literature, the model should either omit precise ratios or cite relevant studies.

Answer:

In our revised manuscript, we added a paragraph and an updated version of Fig. 8, directly showing that FRQ dimers can bind two FRH molecules in HEK293T cells and in *Neurospora* (see responses to all reviewers above).

8. The authors miss many relevant literatures. A part of them is listed below. References (PMID: 38526948, PMID: 24316221) are required when the authors mentioned IDR of FRQ in “The FRQ protein, 85% of which consists of intrinsically disordered regions (IDRs)², dimerizes via a coiled-coil in its N-terminal region 3” or other relevant part.

Answer:

We have cited Hurley et al and Tariq et al., PMID: 24316221, PMID: 38526948,

The references (i.e. PMID: 31554810, PMID: 34182774) are required for “FCD-bound CK1a is sufficient to phosphorylate FRQ on a circadian timescale and in a temperature-insensitive manner, and thus the FFC constitutes a molecular module suited to measure time 16,18.”

Answer:

PMID: 31554810, PMID: 34182774 are included in the revised version.

References are needed here, “For example, CK1a and FRH have essential functions outside the clock and cannot be deleted or inactivated.”

Answer:

We have modified the sentence:

For example, homologues of CK1a and FRH in yeast and mammals have essential functions outside the clock and Neurospora knockouts are not available (Shi et al 2010, PMID: 19948888; Knippschild et al 2014, PMID: 24904820; Liang et al 1996, PMID: 8756671; Pfaffenwimmer et al, PMID: 24968893; Jackson et al 2010, PMID: 20512111).

“FRQ and WC-1 without their binding partners, FRH and WC-2, respectively, are very unstable and thus not expressed at a sufficient level to pursue complex biochemical analyses.”

Answer:

As suggested, we have included references:

FRQ and WC-1 without their binding partners, FRH and WC-2, respectively, are very unstable and thus not expressed at a sufficient level to pursue complex biochemical analyses (Guo 2010; PMID: 20159972; Hurley 2013; PMID: 24316221, Schafmeier 2008; PMID: 19141472)

References (PMID: 12098706, PMID: 12098705) are required here, “In addition, the WCC is a light receptor that resets the circadian clock in response to blue light stimuli.”

Answer

Reference PMID: 12098706, PMID: 12098705 are included in the revised version.

References are required for “In Neurospora, WC-1 is rapidly degraded without its stabilizing partner WC-2 and, hence, cannot be easily analyzed.”

Answer

Reference PMID: 19141472 is included in the revised version.

Minor points :

1. Figure 3C-E, The FRH:FRQ ratio is noted in the legend but not the main text.

Answer:

The high and low FRH to FRQ ratios are noted in the text

2. For Figures S1 and S5, Combine multi-panel figures onto single pages for clarity.

Answer:

Done. Note that we have altered Fig. S1.

3. “It has been reported that specific mutations in the C-terminal third of FRQ affect its interaction with WCC but not with FRH”. “third” or “part” ?

Answer:

Both is right, C-terminal third is more precise. We use “third” in the revised version.

Reviewer #2 (Remarks to the Author):

This manuscript addresses the question of the functional roles of FRQ phosphorylation by

CK1a and FRH binding, two aspects of FRQ function that are clearly important for circadian rhythms in the fungal model system *Neurospora crassa*, but which lack a firmly established mechanistic role. There have been challenges studying CK1a and FRH in *Neurospora* because they are both essential genes. The authors established a heterologous expression system for the fungal proteins in human U2OS osteosarcoma cells, enabling them to look at how FRH binding and CK1a phosphorylation affect nuclear localization and interaction with white collar (WC) proteins, which serve as the central transcriptional regulator of the *Neurospora* circadian clock. They conclude that the well-documented slow hyperphosphorylation of FRQ by CK1a remodels the FRQ-FRH-CK1a (FFC) complex, and make the important discovery that FRH competes with the white collar complex (WCC) and regulates nuclear localization and protein turnover, providing a new model that helps to explain the biochemical basis for the delay in FFC regulation of the WCC in the clock mechanism. Their evidence for this new model is compelling, even if it mostly originates from the heterologous system. Mutations based off the U2OS data are tested in *Neurospora* cycling assays and some in vitro biochemistry provides strong support for the central claim of this work. Overall, this is an exciting piece of work that offers a new explanation for the delay in transcriptional regulation in a circadian clock, leading to many additional studies that can follow up in more detail on the specific sites and mechanisms. However, looking at the figures alone, it is often hard to synthesize the important points, significantly detracting from the impact of the work. There are some comments and suggestions below for improvements that might help convey this information better to readers.

1. It might be helpful to denote graphically which constructs in Figure 1 can interact with FRH. Also, the microscopy images could be smaller and perhaps be accompanied by a graphical summary of the data to assist in interpretation at a glance.

Answer:

All constructs containing the FRQ-FRH binding domain (FFD) including the 6B2 region should interact with FRH. We have indicated this in Figure 1

2. It would be nice to get damping rates for the data in Figure 2.

Answer:

As suggested, we have calculated damping rates (γ), which are indicated in Figure 2.

3. The use of red-green coloring throughout the figures should be changed in consideration of color-blind viewers.

Answer:

We have changed the coloring.

4. It might help to have cartoons of the constructs and interactions alongside the microscopy images in Figure 3, 4, and 6, similar to Figure 5.

Answer:

As suggested by the reviewer, we added cartoons to Figure 3. However, this approach did not work for Figure 4, S5 and 6. We attempted to reduce the size of the microscopic images

and add cartoons, but this made the figure overly busy, cluttered, and confusing. Instead, we have indicated the combination of proteins expressed, e.g.: FRQ + WC-1, ... etc.

5. The data in Fig. S7 is compelling and complementary to the co-IP in Figure 7. I suggest including it in the main figure.

Answer:

We thank the reviewer for the suggestion. In the light that we have already so many main figure panels, we rather leave the gelfiltration analysis in the supplemental figures.

6. It would be nice to have data of FRH NLS mutant(s) to support the claim made in Figure 3e and lines 177-180.

Answer:

While additional insights are always valuable, such experiments would require considerable time and resources, in particular as it is already well established that FRH is nuclear (see, e.g., Cheng et al. 2005 PMID: 15625191, and several other studies).

We demonstrate that FRH facilitates the translocation of FRQ^{NLSmut1/2/3} from the cytosol to the nucleus (Fig. 3A,B). Therefore, we do not consider it a high priority, within the scope of this work, to experimentally identify and mutate the NLS of FRH.

Yet, following the reviewer's comments, we re-examined the NLS predictions. FRH shows three sub-threshold peaks that are absent in the cytosolic MTR4 homologs from yeast and humans. An above-threshold signal is predicted for yeast Ski2; however, since Ski2 is cytosolic, this likely does not represent a true NLS. Consequently, experimentally identifying the NLS(s) of FRH may prove to be labor-intensive. Given the uncertainty and potential for confusion, we chose to delete the NLS prediction for FRH.

Reviewer #3 (Remarks to the Author):

Summary:

This paper explores the molecular events governing FRQ localisation, participation in complexes and stability using combined mutational and microscopy-based approaches. Owing to the intractability of *N. crassa* to live cell microscopy the authors elect to establish a heterologous overexpression model (using U2OS cells) to explore the interplay between FRH, FRQ and phosphorylation (of FRQ) by casein kinase 1. They characterise a series of FRQ mutants in terms of their nuclear/cytoplasmic distribution and relate this to downstream effects on the circadian clock. They later use the same mutants to study both the distribution and colocalization of FRQ with various binding partners.

The findings regarding FRH's regulatory influence on FRQ localization and interactions are of

substantial interest to the wider circadian biology community. However, several methodological and interpretive blind-spots should be addressed before the manuscript is suitable for publication.

Strengths:

This manuscript provides a compelling and novel system for dissecting the spatial regulation and molecular interactions of *Neurospora crassa* clock proteins using a heterologous human cell model. Using this platform, the authors identify sites for regulation of nuclear import and export and demonstrates the role of FRH in modulating localisation. They link the nuclear mis-localisation of different NLS mutants to circadian phenotypes in *N. crassa*. In general, the paper is clearly written and well presented, and it does strongly suggest a novel role for FRH in regulating the interaction of WCC and FRQ.

I am also interested if the authors can rationalise the different NLS mutation phenotypes (Figure2), i.e NLS2 and NLS3 have opposing effects on circadian period, which is an interesting and counter-intuitive (at least to me) observation.

Answer:

The reviewer is correct; this result does appear a bit counterintuitive. Unfortunately, we do not yet understand why the effects on period are opposite. One possible explanation is that individual NLS sequences are inactivated with different kinetics and/or differ in their "import strength". Hence, when only two NLSs are present in FRQ, period length may be determined by their combined import „strength“ relative to the export “strength” of the NES.

Further to this, does introducing the 6B2 mutation to native FRQ abolish rhythmicity, particularly when combined with the triple NLS mutant. Similarly, would a FRQ-NES mutant have a similar circadian phenotype.

Answer:

The 6B2 mutation is arrhythmic; therefore, all combinations involving 6B2 will also be arrhythmic.

Weaknesses:

1. In general, quantification and appropriate statistical tests should be provided for all microscopy data and western blots.

Answer:

Western blots show qualitative data:

Fig 6A shows CK1a-dependent hyperphosphorylation. n =3

Fig 7B shows that CK1a treatment disrupts the Co-IP of FRQ with FRH. n =3

Microscopic images:

Fig 1: The vast majority (if not all) cells expressing the indicated FRQ variants exhibit the subcellular localization shown in the corresponding figure panels.

Fig 3

Same answer, the vast majority (if not all) cells expressing the indicated FRQ variants exhibit the subcellular localization shown in the corresponding figure panels. Specifically, we have evaluated 20 cells per condition.

Fig 4

We have evaluated 20 cells per condition.

A-E: interaction of FRQ with WC-1:

Fig. A,B: 20/20 cells exhibit the subcellular localization shown in the corresponding figure panels.

Fig. 4C: 19/20 cells exhibit the subcellular localization shown in the figure panel.

Fig. 4D is quantified in Fig 5A.

E: 18/20 cells exhibit the subcellular localization shown in the figure panel.

F-I: interaction of FRQ with WCC:

Fig. F-G: 20/20 cells exhibit the subcellular localization shown in the figure panels.

I: 19/20 cells exhibit the subcellular localization shown in the figure panel. i.e. no co-localization of WCC with FRQ⁹.

2. Studying typically low-abundance proteins in the context of over-expression has inherent problems, it is plausible that some of the interactions which are reported here are, in part, a function of their being present at a significantly higher copy number than would be expected in *N. crassa*, though this is caveated by the authors.

In particular, the formation of nuclear aggregates is common consequence of over-expressing transcriptional factors, e.g., Buchberger et al., PLoS One, 2013. Given the size of the foci (several microns) it is plausible that FRQ and FRH may be in nuclear aggregates/inclusions or interacting with other large macromolecular assemblies, RNA, other membrane less organelles or nuclear bodies in a fashion that does not reflect their endogenous function and sub-nuclear localisation. It would be revealing to perform some simple IP or pull-down experiments with affinity-tagged constructs.

Answer:

All clock protein interactions in *Neurospora* were previously identified through pulldown assays of wild-type and mutant proteins conducted by us and others. These interactions are validated/reproduced in our co-localization assays, with specificity confirmed by the absence of co-localization in the respective mutants. Moreover, the co-immunoprecipitation of FRH with FRQ in HEK293T cells (Fig. 8A) provides additional evidence that the co-localizing proteins physically interact.

Furthermore, over-expression artefacts are possibly exacerbated by the combined choice of host (U2OS) and the highly processive CMV-tet promoter. Since the promoter is tet-responsive it should be possible to titrate the expression level with of at least one tagged protein with doxycycline.

Answer:

We have attempted this, but it was unfortunately not possible to titrate expression levels with DOX, as it behaves like an all-or-nothing system. We can only speculate as to the reason. One possibility is DOX metabolism, which may require substantially higher

concentrations to sustain expression over time than are needed to initially induce transcription. Consequently, at low DOX concentrations, genes may be partially induced at first, but as DOX levels fall below a threshold due to metabolism, protein levels do not accumulate to a detectable level. Hence, induction at levels sufficient to sustain detectable protein expression is likely to exhibit switch-like behavior.

Furthermore, where plasmids have been co-transfected one cannot assume equal partitioning of transcriptional and translational machinery, i.e. the observed response when combining different mutants is not just a feature of the biochemical activities of the expressed proteins. Moreover, although fine at the population level it is not reasonable to assume that each dually fluorescent cell has taken up an equimolar ratio of plasmids to that in the transfection mixture unless validated e.g. by calibrating the relative fluorescence intensities.

Answer:

Suitable ratios of co-transfected plasmids were determined empirically. In U2OS cells, individual cells express varying amounts of the proteins. We monitored the cells over time and analyzed only those that expressed both tagged proteins at medium levels (i.e., excluding cells with extremely low or extremely high expression of one or both proteins). The reviewer is of course correct. Detectable interaction depends on ratios, but also on concentrations and binding affinities. For example, when analyzing the interaction of WC-1 with FRQ, we always see colocalization of WC-1 in FRQ nuclear foci but observe in several cases WC-1 remaining in the cytosol, likely due to WC-1 being in excess over FRQ or expression levels not permitting saturation binding to nuclear FRQ. Our primary focus, however, is on whether interaction occurs - e.g., colocalization of WC-1 with nuclear FRQ foci - or whether there is no interaction, such as no colocalization of WC-1 with FRQ⁹ foci.

Since the authors are using a heterologous system, I would have greater confidence in the conclusions of this paper if some microscopy could be recapitulated in at least one other cell-line. This would demonstrate that what is observed here is due to the intrinsic properties of the specific proteins and not cell-line specific.

Answer:

As suggested, we examined protein expression in HEK293T cells. The results (Fig. S2D,E) indicate that the proteins display behavior similar to that observed in U2OS cells.

2. The authors make claims about the overlapping, or not, of WC1 and FRQ foci in Figure 4H-I, respectively. Firstly I do not agree that this is particularly apparent in the figures themselves, nonetheless one would expect some degree of overlap even if they were independently and randomly distributed. Frq-9 signal also appears somewhat more diffuse than Frq-6B2 and so one may make the interpretation it is simply interacting with WC at a lower stoichiometry.

Answer:

The C-terminally truncated FRQ⁹ protein(G663TER) does not interact with WC-1 in *Neurospora*. Truncations beyond aa663 are WCC binding deficient (e.g. Wang et al. 2023 PMID 37220856), which is consistent with our observations, i.e. WC-1 never colocalized with FRQ⁹ foci (comp. Figs. 4A,C):

Proper quantification of the degree of colocalization should be provided using established methods.

The method of quantification for figure 5A is unusual and rather opaque. It appears that cells are classified as exhibiting either nuclear or cytoplasmic localisation of WC-1-mNG, based on the relative fluorescent intensities from each compartment. This approach fails to capture the spectrum of localization, from predominantly nuclear to predominantly cytoplasmic. Nonetheless, since observations are not independent (as one goes up the other goes down), neither are they continuous, therefore it invalidates the choice of statistical test. I would recommend a simple nuclear/cytoplasm ratio be reported instead, this would preserve underlying biological variation and better facilitate comparison between +FRH/-FRH conditions.

Answer:

As suggested, we have quantified in Fig. 5A for 30 cells the nuclear/cytoplasm intensity ratio of WC-1-mNG fluorescence in absence of FRH (WC-1-mNG more nuclear) versus the presence of FRH (WC-1-mNG more cytosolic).

3. The authors make multiple allusions to the ability of FRQ to homodimerize and form the basis of various intermediate complexes. Though reduced foci formation by smaller fragments is consistent with a role of the entire molecule forming intermolecular multivalent interactions it doesn't necessarily prove that foci formation is caused by a biologically relevant assembly process (phase-separation or otherwise), nor does it necessarily mean they are functionally important for clock output. Some FRAP experiments on the various FRQ foci would go some way to demonstrating this.

Answer:

The molecular basis of foci formation and its physiological relevance are important questions, but they are beyond the scope of this study and therefore not the focus of our investigations. In the present work, we use foci formation merely as a convenient tool to study colocalization (using the appropriate controls/mutants to avoid artifacts). Nevertheless, we are currently exploring this topic and have already made some intriguing, unexpected and exciting preliminary observations.

Regarding figure 6, the blot lacks quantification and loading control, without this one cannot claim that endogenous CK1 or other kinases are not phosphorylating FRQ.

To aid the comparison and draw a parallel with the blot, it would be helpful to have some live cell microscopy without CK1a overexpression.

Answer:

In U2OStx cells FRQ, as well as FRQ^{6B2} and FRQ⁹, reach over time a higher phosphorylation state in cells co-expressing CK1a than in cells that do not express CK1a. This does not exclude that endogenous kinases do not phosphorylate FRQ to some (minor) extent.

We changed the wording in the main text:

This indicates that CK1a phosphorylated mK2-FRQ in U2OStx cells while endogenous kinases did not support similar hyperphosphorylation of the overexpressed FRQ proteins.

However, potential phosphorylation by endogenous U2OS kinases does not impact FRQ export to the cytosol.

As suggested, we therefore include (as Fig. S5B) a time course showing that FRQ without CK1a remains nuclear at all times:

4. Though the model the authors propose (Figure 8) is certainly plausible, it is not clear to me what is the direct evidence for the existence of the specific intermediates in the data presented (i.e gradual dissociation of FRH), which is surprising given the suggested slow phosphorylation kinetics. Additional experiments (ie. Native-PAGE/crosslinking etc.) would be required to interrogate the relative stoichiometries of the complex participants.

Answer:

We show pulldown from HEK293T cells and *Neurospora*, demonstrating that a FRQ dimer can bind two FRH molecules (see answer to all reviewers above).

Concerning the stepwise release, and assuming that the CK1a-dependent dissociation of the two FRH molecules from the FRQ dimer is not strictly coupled but occurs independently, we provide here a theoretical calculation to illustrate the effect of slow phosphorylation on FRH release:

Assume a FRQ dimer initially binds two FRH molecules. If slow phosphorylation by CK1 triggers FRH release at a rate of e.g. 0.25 h^{-1} ($t_{1/2} \approx 2.8 \text{ h}$, median), the average waiting time (mean) for dissociation of the first FRH - producing an FRQ_2FRH_1 species - equals $t = 1/(2 \times 0.25 \text{ h}^{-1}) = 2 \text{ h}$.

The second FRH dissociates after an additional average waiting time of $1/(0.25 \text{ h}^{-1}) = 4 \text{ h}$. Consequently, in this theoretical example, the average time delay before inactive FRQ_2FRH_2 becomes active via release of the first FRH is 2 h, and the resulting time window during which species with only one FRH bound (FRQ_2FRH_1) remain active is 4 h.

If FRQ were produced at a steady rate, the ratio of FRQ_2FRH_2 to FRQ_2FRH_1 would be about 1:2. In a physiological context (*Neurospora*), where negative feedback operates to shut down FRQ synthesis, FRQ_2FRH_2 would be converted to FRQ_2FRH_1 , which would become the predominant FRQ species (FRQ_2 is degraded and does not accumulate). The experimental

observation that FRQ₂FRH₂ are detected in HEK293T cells and with the expected low abundance in *Neurospora* (see new Fig. 8) is consistent with the stepwise release model.

Minor point: The Y-axes of Figure 2 are not consistent (‘.’ and ‘,’). It would be helpful to add the amplitude and damping rates if they intend to comment on these in the text.

Answer: Done.

RESPONSE TO REVIEWER COMMENTS

We thank all reviewers for the continued engagement with our manuscript.

Reviewer #1 (Remarks to the Author):

The authors have addressed my concerns.

We thank the Reviewer

Reviewer #2 (Remarks to the Author):

The authors have done an acceptable job addressing my prior comments. The addition of the cartoons makes it easier to move between the text and figures to process the information. There are still many questions remaining that will require future studies to dissect, but given that this type of biochemical study is not possible in the fungal system, I think there is value in proposing this new model. Although the conclusions here depend on overexpression in a heterologous system, I am satisfied with the biochemical data supporting the conclusions.

I do have two minor comments that should be easy to address.

The new co-IP data in figure 8 should state in the figure what protein is being precipitated. This is done in figure 7 and it is standard practice.

Done.

The cartoon present in figure 8 has different coloring than the ones used throughout the other figures. It may make sense to unify the coloring here.

Color has been changed

Reviewer #3 (Remarks to the Author):

In the revised manuscript the authors have made a comprehensive attempt to address several important points. However, some methodological and interpretative concerns remain about the steps the authors have taken to mitigate the risk of confirmation bias, because standard practices such as loading controls, quantification, statistical analyses are still not consistently employed throughout.

More generally the authors are of course entitled to their opinions, and presumably find their interpretation very convincing. Whilst I find their interpretations plausible, I do not personally find the supporting evidence even comes close to being unambiguous. If the authors are reluctant to do further experiments and analyses then I would encourage them to tone down the confidence with which they present their interpretation. Starting with the abstract, I suggest that they should be clear to distinguish their interesting experimental observations from their informed speculations.

We have softened the wording in the abstract: ... We present data *suggesting* that FRH ...

More detailed responses are below:

Reviewer response:

Despite the qualitative nature of the data, loading controls are still recommended to ensure equal loading/transfer and not addressed in the methods. Tubulin loading controls are shown for supplementary figure 1. It is not clear why they are not shown for other blots.

We show Ponceau S staining of the membranes as loading controls in the supplementary data.

Reviewer response:

Using established metrics of co-localization would add rigor to this study and add substance to the claims made in the text. If 20 cells were evaluated then why not show the quantification of the evaluation.

We have quantified the images as requested:

Fig 3: Co-localization of FRH-mNG with mK2-FRQ foci was quantified by evaluation of foci in 20 cells for each FRQ variant. The quantitative results are presented in Supplementary Table 1.

Fig. 4: We have measured the enrichment/density of WC-1-mNG (-/+ untagged WC-2) in nuclear foci of mK2-FRQ (4A, D), mK2-FRQ^{6B2} (4B, E) and mK2-FRQ⁹ (4C, F).

To accommodate the new quantification, Fig. 4 was reorganized. The co-localization of mK2-WC-2 with FRQ-mNG has been moved to Supplementary Figure 2D. The co-localization of WC-1-mNG with the mK2-FRQ C-terminal construct is shown and quantified in Figure 5A. The main text was revised accordingly.

Reviewer response:

Although evidence from *Neurospora* is supportive of the author's general thesis, for the previously mentioned reasons, foci formation in of itself does not provide sufficient proof that localization occurs through the specific interactions mentioned. To establish U2OS overexpression as a useful system for studying mechanisms of *Neurospora* protein complexes there is an obligation to demonstrate that foci are not artefactual. The IP presented in figure 8 is a step towards this but it is unclear why this was done only in HEK293 and not also in U2OS.

The IP was done in HEK293T cells because the transfection efficiency of U2OStx cells is low.

It is well established that FRQ interacts with FRH, whereas the mutant versions FRQ6B2 and FRQ9 do not. It is also firmly demonstrated that FRQ interacts with WCC, while the FRQ9 mutant fails to do so.

Our co-localization approach faithfully reproduces these known interactions and the impact of mutations, validating our approach. Therefore, we find no basis for the reviewer's concern that the observed co-localization could represent an artifact in U2OS cells. It is difficult to see what kind of artifact the reviewer has in mind that could precisely reproduce the established interactions and mutant phenotypes and subcellular localization and still be regarded as artificial?

Moreover, it is very well to say that cells are selected based on a medium level of expression, but it is not clear from the text how this is determined.

Medium expression levels correspond to those shown in the figure panels. We use time-lapse microscopy, which allows us to select cells that neither express at very low levels (predominantly at early time points) nor at excessively high levels (predominantly at late time points). Evaluation of both extremes do not yield different results but are less suitable for clear visual presentation in the figures.

Reviewer response:

The point I intended to make is that Figure H and Figure I appear similar. How this interaction occurs in *Neurospora* is somewhat irrelevant. The onus is on the authors to demonstrate this system faithfully reflects endogenous interactions. Without proper quantification of the degree of localization, the reader is in the position of having to take the authors explanations of the images in good faith.

As requested, we have now quantified the data in Fig. 4 as answered above. Again, I want to stress that all FRQ or FRQ6B2 foci contained WC-1 while we did not find FRQ9 foci containing WC-1.

Reviewer response/Figure 8:

The figure legend and main text appear to suggest different experimental set-ups.

Main text:

“FLAG-FRH and mK2-FRH were co-expressed with, or without mK2-FRQ”

In any case, I found this figure and the accompanying text confusing. The way the blots are presented is not helpful. Some of the blots are overloaded, and it is not clear what the arrows are pointing to in the upper panel of 8a.

We apologize for the mix-up. In the revised version, we replaced the IP and adapted the text and figure legend accordingly. We also present a schematic illustrating the rationale of the CoIP and include two types of negative control FLAG-IPs:

1. **Omitting FLAG-FRH:** No co-IP of mK2-FRH and mK2-FRQ.
2. **Omitting mK2-FRQ:** No co-IP of mK2-FRH with FLAG-FRH.

Critically, this experiment is not suitable to infer complex stoichiometry, SDS-PAGE is a bulk measurement and cannot definitively establish the presence of said complexes.

8b/c similarly to 8a is not suitable to infer stoichiometry. The combined experiments only weakly support the suggested model.

The question we sought to address was whether a FRQ dimer can bind two molecules of FRH. To test this, we used two immunologically distinguishable versions of FRH. The rationale behind the Co-IPs shown in Fig. 8 is that if a FRQ dimer can indeed bind two FRH molecules, it should be possible to co-precipitate one FRH variant via the specific tag of the second. This type of approach is widely used.

In panel 8A (HEK cells), mK2-FRH is detected in an IP of FLAG-FRH (FLAG-IP). The co-IP is dependent on (mediated by) mK2-FRQ.

In panels 8B and 8C (*Neurospora*), endogenous untagged FRH is detected in an IP of GFP-FRH (GFP-pulldown). The co-IP is dependent on (mediated by) FRQ.

Together, these results establish that complexes with 2:2 (FRQ₂FRH₂) stoichiometry exist. See also answer above.

“To directly examine the subunit composition of FRQ-FRH complexes in the absence of CK1a, we expressed the proteins in HEK293T cells”. This statement is not strictly true, HEK293T cells also possess endogenous CK1a.

HEK cells do not express CK1a. They endogenously express CK1δ, which does not phosphorylate FRQ to a detectable extent and does not support dissociation of FRQ-FRH complexes.